# Less is More: an Attention-free Sequence Prediction Modeling for Offline Embodied Learning

**Wei Huang**[1,2][*]   **Jianshu Zhang**[1,3]   **Leiyu Wang**[3]   **Heyue Li**[4]   **Luoyi Fan**[3]
**Yichen Zhu**[5]   **Nanyang Ye**[3][†]   **Qinying Gu**[1][†]

[1]Shanghai AI Laboratory    [2]Tsinghua University    [3]Shanghai Jiao Tong University
[4]Wuhan University    [5]Midea Group
{huangwei, guqinying}@pjlab.org.cn
{jianshuzhang, leiyuwang33, luoyi_fan,ynylincoln}@sjtu.edu.cn
liheyue@whu.edu.cn   yichen_zhu@foxmail.com

## Abstract

Offline reinforcement learning (offline RL) is increasingly approached as a sequence modeling task, with methods leveraging advanced architectures like Transformers to capture trajectory dependencies. Despite significant progress, the mechanisms underlying their effectiveness and limitations remain insufficiently understood. We conduct a thorough analysis on the representative Decision Transformer (DT) model using an entropy analysis and identify the inconsistencies in state-action-reward ($\langle s, a, R \rangle$) distributions causing attention "dispersal". To address this, we propose a hierarchical framework that decomposes sequence modeling into intra-step relational modeling—handled by a Token Merger that fuses each $\langle s, a, R \rangle$ triplet—and inter-step modeling—handled by a Token Mixer across timesteps. We investigate several Token Merger designs and validate their effectiveness across various offline RL methods. Furthermore, our theoretical analysis and experimental results suggest that while Token Mixers are important, lightweight architecture can also achieve even better performance to more complex ones. We therefore propose a parameter-free Average Pooling Token Mixer, which, combined with a convolutional Token Merger, forms our final model, Decision HiFormer (DHi). DHi achieves a **73.6%** improvement in inference speed and an **9.3%** gain in policy performance on the D4RL benchmark compared to DT. DHi also generalizes well to real-world robotic manipulation tasks, offering both practical benefits and insights into sequence-based policy design for offline RL. Code and models are public at project page.

## 1 Introduction

Offline Reinforcement Learning (offline RL) focuses on learning policies from fixed, static datasets without further environment interaction. Traditional methods often rely on policy regularization or value function approximation. Recently, a shift has occurred toward framing offline RL as a sequence prediction problem, where policies are learned by predicting actions based on sequences of past state-action-reward triplets. This reformulation enables the use of advanced sequence modeling techniques, particularly various token mixers, to capture dependencies within trajectories.

A representative work is Decision Transformer (DT) [6] which leverages Transformers [29] to model entire trajectories through global self-attention and has inspired numerous follow-up methods [4,

---

[*]This work was done when Wei Huang interned at Shanghai Artificial Intelligence Laboratory.
[†]Qinying Gu and Nanyang Ye are co-corresponding authors.

39th Conference on Neural Information Processing Systems (NeurIPS 2025).

5, 10, 12, 14, 15, 18, 20–22, 25–27, 30–33]. Despite its great success in this domain, some studies highlight its limitations in local dependency modeling. To address it, the Decision ConvFormer (DC) [15] replaces self-attention with convolutional layers to enhance local feature extraction. StARformer [25] adheres to the self-attention mechanism by explicitly separating short-term and long-term dependencies through interleaved Step and Sequence Transformers, enhancing both local and global trajectory modeling. Similarly, Graph Decision Transformer (GDT) [12] uses a graph transformer to capture potential dependencies between adjacent states, actions, and rewards and differentiate the impact of these different tokens.

The aforementioned previous studies demonstrate that the standard DT architecture is not well-suited for modeling local dependencies in a Markov process [15, 22, 27]. This is because the DT tends to consider all past context preventing it from effectively "attending" to useful states. To mitigate this, alternative operators such as convolutions, linear attention, and state-space models are proposed to replace the self-attention layers for improved performance. However, several key questions remain unresolved: (1) Why is the attention mechanism in DT unable to "attend" to useful states in the context? (2) Beyond performance scores, can we measure the model's ability to capture local dependencies using internal metrics? (3) Does offline RL exhibit significant differences across various operators, and are there simpler, more effective operators available?

To address the above questions, we make the following contributions: (1) We introduce *attention entropy* as a novel metric to theoretically explain the limitations of DT in modeling local dependencies, attributing attention dispersion to inconsistent state-action-reward distributions; (2) We propose a *Token Merger* to harmonize intra-step representations through explicit modality alignment, demonstrating universal effectiveness across different architectures for enhanced trajectory modeling; (3) We develop a parameter-free *Token Mixer* using windowed average pooling [34] to capture inter-step dependencies and preserve the Markov property, validated through theoretical analysis and experiments for efficiency and generalization; (4) We present a generalized hierarchical framework that decomposes sequence modeling into intra-step modeling (*Convolution-based Token Merger*) and inter-step modeling (*Pooling Token Mixer*) operations, achieving a 73.6% increase in speed and an 9.3% improvement in performance on the D4RL benchmark compared to DT, as shown in Figure 1; (5) We demonstrate the generalization of our method to real-world robotic arm operations, achieving a high success rate, smooth trajectory execution, and fast inference speed.

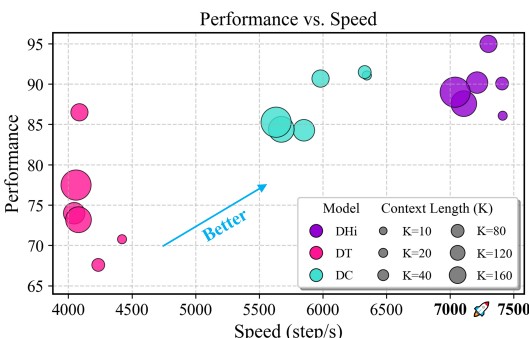

Figure 1: Performance and speed of the proposed Decision HiFormer (DHi) are compared with DT and the recently proposed DC on the MuJoCo Hopper-medium task. The results show that our method achieves significant improvements in both performance and speed.

## 2 Related Work

### 2.1 Offline RL

Offline reinforcement learning (offline RL) learns policies from fixed datasets without environment interaction, but suffers from distributional shift between the learned and behavior policies [19]. Q-learning method is one of the most prominent categories to address this by learning conservative or constrained value functions [10, 11, 16, 17, 28]. For example, Batch-Constrained Q-learning (BCQ) [11] constrains the learned policy to remain near the dataset support by using a generative model and action perturbation. Conservative Q-learning (CQL) [17] penalizes Q-values for actions not in the datasets, explicitly enforcing conservatism. Implicit Q-learning (IQL) [16] avoids behavior constraints by decoupling policy and value learning, using advantage-weighted regression. While effective, these methods rely on value estimation and are sensitive to bootstrapping error.

## 2.2 Sequence Modeling for Offline RL

Recent advances in offline RL have shifted focus toward sequence modeling[6, 14], which reframe policy learning as predicting future actions from historical trajectories using architectures that capture temporal dependencies. A pioneering work, DT [6], uses Transformers to model long-term dependencies by conditioning actions on past states and future returns, eliminating the need for bootstrapping. DT's success inspires various follow-ups leveraging Transformers for temporal feature modeling. For example, EDT [31] introduces a dynamic history length mechanism to address DT's trajectory stitching limitation, while Waypoint Transformer [2] focuses on implicit goal generation for trajectory planning. GDT [12] combines Graph Neural Networks (GNN) with Transformers to model complex state-action relationships. StARformer [25] uses interleaved Step and Sequence Transformers to explicitly separate short-term and long-term dependencies.

Despite their success, vanilla self-attention's global focus can overlook local patterns and incurs significant computational overhead. To alleviate these issues, alternative lightweight structures have been proposed. For instance, RvS [8] uses a two-layer MLP to condition on current state and return; DC [15] replaces self-attention with local convolution to enhance local modeling; and FCNet [27] adopts Fourier transforms to capture low-frequency trajectory patterns for real-time decision-making. State-space Models (SSMs), such as S4 [3] and Mamba [7, 22] have also been adopted as lightweight alternatives to self-attention, offering enhanced local dependency modeling.

# 3 Method

## 3.1 Preliminaries: Trajectory Modeling and Attention Structure

In offline reinforcement learning (RL), the goal is to learn a policy that maximizes the expected cumulative return from a fixed dataset of trajectories, without any further environment interaction. Each trajectory $\tau = (s_0, a_0, r_0, \ldots, s_L, a_L, r_L)$ is collected under an unknown behavior policy, where $s_t$, $a_t$, and $r_t$ denote the state, action, and reward at timestep $t$, respectively. To enable autoregressive modeling, a common approach—exemplified by DT—represents trajectories as sequences of return-to-go (RTG), state, and action tokens.

For each timestep $t$, a subtrajectory of length $K$ is constructed in the form:

$$\tau_{t-K+1:t} = (\hat{R}_{t-K+1}, s_{t-K+1}, a_{t-K+1}, \ldots, \hat{R}_t, s_t), \tag{1}$$

where the return-to-go is defined as $\hat{R}_t = \sum_{t'=t}^{L} r_{t'}$. Each token in the subtrajectory is embedded into a $d$-dimensional vector and passed through a multi-layer Transformer encoder.

Given an input of length $N = 3K$, multi-head self-attention is applied. For each attention head $h \in \{1, \ldots, H\}$, attention weights are computed using linearly projected queries and keys:

$$\alpha_{ij}^h = \text{softmax}\left(\frac{(x_i W_Q^h)^\top (x_j W_K^h)}{\sqrt{d_k}}\right), \tag{2}$$

where $x_i \in \mathbb{R}^d$ is the embedded token at position $i$, and $W_Q^h, W_K^h \in \mathbb{R}^{d \times d_k}$ are learnable projection matrices. These attention weights form the basis for downstream value aggregation and temporal dependency modeling.

## 3.2 Quantifying Attention Focus with Attention Entropy

**Mathematical Formulation**  Although Transformers possess strong sequence modeling capabilities, their attention mechanism is inherently position-agnostic and does not favor temporally local structure. In offline RL, however, effective decision-making often relies on short-range temporal dependencies—e.g., transitions from $t - 1$ or $t - 2$—due to the Markovian nature of the environment. These short-range inter-step relationships, referred to here as *local dependencies*, play a crucial role in credit assignment and behavioral consistency across trajectories. To evaluate whether the model captures such local dependencies, we analyze the sharpness of its attention distributions. Specifically, we adopt *attention entropy* as a proxy measure. Let $\alpha_{ij}^h$ denote the attention weight from token $i$ to token $j$ in head $h$ of the Transformer. These weights can be interpreted as a probability distribution

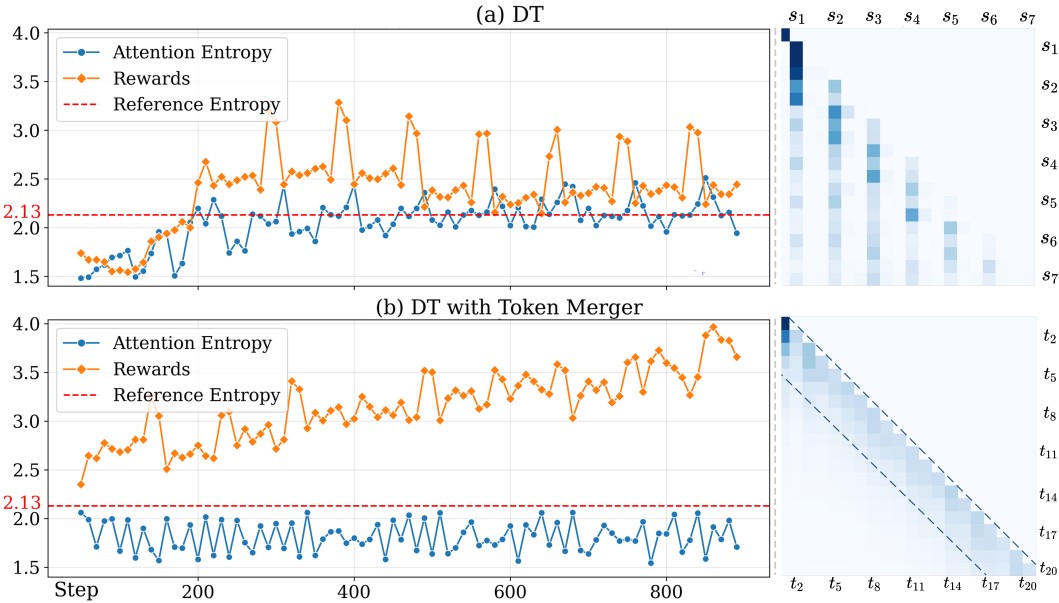

Figure 2: Entropy-reward correlation (left) and attention maps (right) for DT (top) and DT with Token Merger (bottom), where lower entropy signifies more focused attention and thus stronger local dependency modeling. DT exhibits state-dominated attention, with sparse weights on RTG and action tokens, leading to higher attention entropy and a strong correlation between local entropy maxima and reward minima. In contrast, DT with Token Merger achieves significantly lower entropy, more balanced inter-step dependencies, and notably higher rewards.

over keys $j$ for each query $i$, conditioned on the attention head $h$. The average attention entropy for head $h$ is defined as:

$$S_h = -\frac{1}{N} \sum_{i=1}^{N} \sum_{j=1}^{N} \alpha_{ij}^h \log \alpha_{ij}^h, \tag{3}$$

where $N = 3K$ is the total number of input tokens in the subtrajectory (with each timestep contributing RTG, state, and action tokens). The overall attention entropy is obtained by averaging across all $H$ heads: $S = \frac{1}{H} \sum_{h=1}^{H} S_h$. A lower value of $S$ indicates more focused attention distributions, often associated with stronger local dependency modeling, while high entropy implies dispersed, possibly uninformative focus.

**Empirical results** We empirically evaluate this relationship in the MuJoCo Hopper-medium environment by measuring attention entropy and its correlation with rewards from the first layer of DT after a warm-up of 50 steps. As shown in Figure 2(a, left), a clear negative correlation emerges: spikes in entropy align with sharp reward drops, while periods of low entropy coincide with improved returns. Notably, low entropy does not always guarantee high reward—likely due to delayed credit assignment—but high entropy consistently correlates with degraded performance. These patterns indicate that attention dispersion undermines stable decision behavior.

We further disentangle intra-step and inter-step behaviors by analyzing attention maps of DT from the perspective of attention entropy (Figure 2(a, right)). Within a timestep, attention is highly concentrated on state tokens, with RTG and action tokens receiving minimal weight. We refer to the attention entropy within a single time step as *intra-step modeling entropy*. This modality imbalance indicates that DT's intra-step attention is focused but biased—emphasizing state over other decision-relevant inputs. When extending this analysis across multiple timesteps, however, the attention scores for the dominant state become more dispersed across time steps, forming a full lower triangular matrix, aligning with the observations in [15]. We treat the state, action, and RTG of a single time step as a whole and refer to the attention score entropy across different time steps as *inter-step modeling entropy*. The inter-step modeling entropy is consistently higher, indicating that while DT exhibits structured intra-step attention behavior, it struggles to capture inter-step dependencies. This

observation supports a key insight: DT's local modeling limitations stem not from weak intra-step fusion, but from its inability to propagate relevant information across adjacent timesteps. Note that in this context, local dependency modeling does not refer to intra-step modeling but the modeling between very adjacent timesteps.

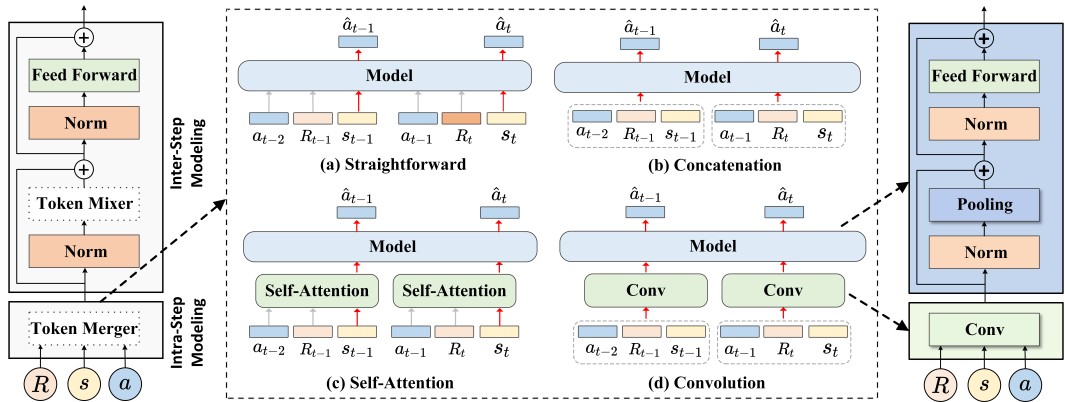

Figure 3: Hierarchical modeling framework and micro structure of Token Merger. Token Merger merges $< R, s, a >$ triplets into unified representations, while Token Mixer models dependencies across timesteps.

## 3.3 Structured Entropy Reduction: Token Merger

To address the limitations in inter-step modeling observed in DT, a natural idea would be to train the model via backpropagation to capture dependencies across timesteps. However, since Transformers model pairwise interactions between individual tokens, learning relationships between coherent $\langle s, a, R \rangle$ triplets becomes challenging. The standard DT struggles with this due to the absence of timestep-level structure in its token sequence. Our proposed solution is to first merge the components of each $\langle s, a, R \rangle$ triplet into a single composite token, and then model dependencies across these merged tokens. To this end, we introduce two modules: the *Token Merger* for intra-step modeling and the *Token Mixer* for inter-step modeling.

The *Token Merger* merges the state, action, and RTG within each timestep into a single unified representation, dynamically balancing the contributions of each component to prevent Transformer from over-relying on state while neglecting action and return-to-go. This design ensures that decision-making accounts for the dynamic relationship between state transitions and reward optimization. Additionally, in its original form, DT was responsible for both intra-step and inter-step modeling, but it struggled to balance these tasks. The *Token Merger* relieves this burden by providing timestep-level representations, allowing the Transformer to focus exclusively on inter-step modeling. Assuming that the total attention entropy remains unchanged after adding the Token Merger, we can apply Theorem 3.1. to conclude that DT's overall attention entropy should decrease.

**Theorem 3.1.** *Let $P = (p_1, \ldots, p_k)$ represent the original attention probability distribution. By grouping tokens into triplets: $G_i = \{p_{3i-2}, p_{3i-1}, p_{3i}\}$, $P' = (g_1, g_2, \ldots)$, $g_i = \sum_{j \in G_i} p_j$. The entropy can be decomposed using the chain rule:* $H(P) = \underbrace{H(P')}_{\text{Inter-group}} + \underbrace{\sum_i g_i H(G_i)}_{\text{Intra-group}}$, $H(G_i) = -\sum_{j \in G_i} \frac{p_j}{g_i} \log \frac{p_j}{g_i}$ *Since $H(G_i) \geq 0$, we can conclude:* $H(P) \geq H(P')$.

**Remark:** This inequality guarantees that triplet grouping reduces attention entropy, enabling more focused attention across timesteps.

We further analyze the attention entropy and attention map of DT with a Token Merger in the same Hopper-medium environment discussed in Sec.3.2. As shown in Figure 2(b, left), introducing a Token Merger significantly reduces average attention entropy and yields a more stable and higher reward trajectory, supporting the conclusion of Theorem 3.1.. In terms of attention distribution (Figure 2(b, right)), compared to the original DT model—which exhibits modality imbalance within

Table 1: Performance in the Gym domain with different Token Mixer and Merger configurations. All models are trained for 20k steps, and scores are averaged over nine task-dataset pairs (combinations of Halfcheetah, Hopper, Walker2d tasks and medium-expert, medium, medium-replay datasets) and five random seeds.

| Mixer \ Merger | Straightforward | Concatenation | Attention | Convolution | Pooling | **Average** |
|---|---|---|---|---|---|---|
| None (MLP only) | 60.7 | 75.6 | 79.8 | 79.3 | 75.5 | 74.2 |
| Linear Attention | 74.6 | 78.1 | 78.8 | 80.4 | 80.0 | **78.4** |
| Attention | 73.5 | 76.9 | 76.4 | 78.9 | 75.4 | 76.2 |
| Convolution | 76.7 | 78.6 | 76.4 | 80.1 | 79.8 | 78.3 |
| Pooling | 75.0 | 78.5 | 78.5 | 80.7 | 78.1 | 78.2 |
| **Average** | 72.1 | 77.5 | 78.0 | **79.9** | 77.8 | 77.1 |

each timestep—the merged attention pattern is more structured and focused across timestep-level tokens. This suggests that the Token Merger not only compresses intra-step information but also promotes a more focused and temporally coherent inter-step attention profile.

## 3.4 Hierarchical Modeling Framework

This subsection outlines our hierarchical trajectory modeling framework. We first define the *Token Merger* for intra-step fusion and evaluate several merger operations, ultimately selecting convolution for its effectiveness. We then analyze the role of the *Token Mixer* in inter-step modeling, examining its necessity, architectural choices, and interaction with the Token Merger. Motivated by the short-range nature of dependencies in offline RL, we demonstrate through both empirical results and theoretical analysis that a lightweight, non-parametric average pooling operation is an even preferable *Token Mixer* to more complex alternatives. Together, these components form our final model, shown in Figure 3.

**Token Merger.** As mentioned, Token Merger maps each timestep's triplet $(\hat{R}, s, a)$ into a unified representation $f : \mathbb{R}^{3 \times d} \to \mathbb{R}^d$. To ensure the alignment of $\hat{R}$, $s$ and $a$ in the sequence, we obtain the input by performing an action shift to the trajectory $\tau_{t-K+1:t}$, restructuring it as: $\tau'_{t-K+1:t} = \left( a_{t-K}, \hat{R}_{t-K+1}, s_{t-K+1}, a_{t-K+1}, \ldots, \hat{R}_t, s_t \right)$. This adjustment ensures that the model predicts $a_t$ using $(a_{t-1}, s_t, R_t)$ rather than $(s_t, R_t, a_t)$, reinforcing a structured input format.

In standard sequence modeling architectures for offline RL—comprising an embedding layer, encoder blocks, and a prediction head—the Token Merger is applied to trajectory sequences before they enter the encoder blocks (Figure 3). We evaluate four types of single-step Token Mergers: Concatenation directly merges the state, action, and return vectors by combining them before applying an embedding layer; Self-Attention introduces a local attention mechanism over adjacent tokens to capture short-range dependencies [25]; Convolution applies a kernel to extract local patterns from neighboring tokens; Pooling performs an average operation over adjacent tokens to aggregate their representations into a single merged token. Experiments in the Gym domain show that performance is lowest when no merger is used, and highest with convolution (Table 1). Our experimental results are consistent with our intuition that convolution performs the best because it effectively captures local dependencies between the tokens and smooths the information fusion.

**Token Mixer.** As mentioned, recent studies have explored various token mixers, such as self-attention [6], linear attention [27] and convolution [15], all of which have demonstrated strong performance in offline RL tasks. This raises three key questions: (1) Is the Token Mixer essential for effective offline RL? (2) Do different Token Mixers significantly impact performance? and (3) Can the Token Merger mechanism generalize across different types of Token Mixers?

We evaluate various token mixers on nine Gym task-dataset pairs, as shown in Table 1. The experimental results indicate that: (1) The Token Mixer is essential, particularly in the absence of a Token Merger. Models without a Token Mixer (i.e., MLP only) perform significantly worse than those incorporating Token Mixers. However, with the Token Merger in place, the Residual MLP (without a Token Mixer) already performs reasonably well, similar to behavior cloning (BC) methods that cannot utilize historical trajectory information or cumulative returns. (2) There is limited

performance variation between different token mixers. (3) Most Token Mixers achieve substantially better performance when combined with a Token Merger, demonstrating that the effectiveness of the Token Merger extends across different Token Mixers.

**Hierarchical Framework with Token Merger and Token Mixer.** Based on these observations, we propose a hierarchical framework that integrates both a Token Merger and a Token Mixer. While the Token Merger alone, when combined with residual MLP, can implicitly model temporal dependencies, it struggles with stability across timesteps due to the lack of context. Meanwhile, Transformer-style Token Mixers incur high computational costs and do not necessarily improve across-timestep dependency modeling, often overemphasizing long-range interactions that are less relevant in offline RL. Given that decision-making in offline RL is primarily driven by short-range dependencies, and that excessive modeling capacity can lead to instability or inefficiency, we propose a non-parametric average pooling operation as the Token Mixer. This aggregates adjacent timestep representations, providing stable cross-timestep smoothing without introducing additional parameters.

The convolution-based Token Merger encodes each token as an $\langle R, s, a \rangle$ tuple, reducing the input sequence length to one-third of its original size and improving the computational efficiency of the multi-layer encoder. Given this reduction, a pooling size of 2 is sufficient to capture temporal information from the current and preceding timesteps. We provide a theoretical proof of the pooling layer's expressive power in modeling Markov problems in Appendix A and Appendix B. We refer to our model as **Decision HiFormer (DHi)**, a lightweight, hierarchical, and high-performance alternative to Transformer-based architectures that demonstrates strong empirical and theoretical performance in modeling local dependencies.

# 4 Experiment

In this section, we present a comprehensive evaluation of the model using the widely recognized D4RL benchmark [9]. A broad range of offline RL baselines are included. To bridge the gap between simulated benchmarks and real-world applications, we further deploy DHi on a 7-DOF Franka Emika Panda robotic arm for physical validation. Ablation studies are also conducted to examine the impact of different window sizes in the pooling layer and different Token Mergers on model performance. We further demonstrate the outstanding performance of DHi in sparse reward environments as well as its generalization across context lengths.

## 4.1 Experiment Setting

**Baselines** We compare our approach with existing SOTA offline RL approaches, each excelling in specific domain tasks. For value-based approaches, we include IQL [16], BCQ [11], and CQL [17]. For supervised learning approaches, we include DT [6], StAR [25], GDT [12], FCNet [27] and DC [15]. Further experimental details and more baseline comparisons are provided in Appendix C and Appendix D, respectively.

**D4RL Datasets** We explore four domains in the D4RL benchmark, each presenting unique challenges for evaluating offline RL algorithms. The Gym-MuJoCo environments provide a straightforward assessment platform with near-optimal trajectories and clear reward structures. In contrast, the Adroit domain, based on human demonstrations, features a constrained state-action space where the agent must maintain reasonable behavior within strict limits. The Kitchen domain involves multi-task complexity, requiring agents to complete four sequential subtasks in a specific order and generalize beyond training data to reach the target configuration. Finally, the AntMaze domain uses an 8-DOF robotic agent with sparse rewards and increased locomotor complexity to make navigation even more difficult.

**Real-World Environment Datasets.** We further evaluate the model's inference speed and generalization ability in real-world settings using a 7-DOF Franka Emika Panda robotic arm equipped with a single-joint torque sensor. The experiments involve four tasks of varying complexity, as shown in Figure 4. In the *Lift task*, the robotic arm lifts a green cube, while in the *Place task*, it picks up a teddy bear and places it into a green bowl. The *Selective Lift* and *Selective Place* tasks introduce additional challenges, with *Selective Place* incorporating obstacles to assess the model's

Table 2: Performance of DHi and baselines on Gym, Adroit, Kitchen and AntMaze domains. All models are trained for 100k steps, and scores are averaged over five random seeds. Best and second-best scores are denoted in bold and underlined, respectively.

| Gym Tasks | CQL | IQL | BCQ | BC | DT | StAR | GDT | FCNet | DC | Ours |
|---|---|---|---|---|---|---|---|---|---|---|
| halfcheetah-medium-expert | 91.6 | 86.7 | 69.6 | 55.2 | 86.8 | 93.7 | 93.2 | 91.2 | 93.0 | **94.2** |
| hopper-medium-expert | 105.4 | 91.5 | 109.1 | 52.5 | 107.6 | 111.1 | 111.1 | 110.5 | 110.4 | **111.7** |
| walker2d-medium-expert | 108.8 | **109.6** | 67.3 | 107.5 | 108.1 | 109.0 | 107.7 | 108.8 | **109.6** | **109.6** |
| halfcheetah-medium | **49.2** | 47.4 | 41.5 | 42.6 | 42.6 | 42.9 | 42.9 | 42.9 | 43.0 | 43.4 |
| hopper-medium | 69.4 | 66.3 | 65.1 | 52.9 | 67.6 | 59.5 | 77.6 | 57.8 | **92.5** | 90.1 |
| walker2d-medium | **83.0** | 78.3 | 52.0 | 75.3 | 74.0 | 73.8 | 76.5 | 75.2 | 79.2 | 79.9 |
| halfcheetah-medium-replay | **45.5** | 44.2 | 34.8 | 36.6 | 36.6 | 36.8 | 40.5 | 39.8 | 41.3 | 41.5 |
| hopper-medium-replay | 95.0 | 94.7 | 31.1 | 18.1 | 82.7 | 29.2 | 85.3 | 85.8 | 94.2 | **97.7** |
| walker2d-medium-replay | 77.2 | 73.9 | 13.7 | 32.3 | 79.4 | 39.8 | 77.5 | 63.5 | 76.6 | **81.2** |
| **Average** | 80.6 | 77.0 | 53.8 | 52.6 | 76.2 | 66.2 | 79.1 | 75.1 | 82.2 | **83.3** |
| **Adroit Tasks** | CQL | IQL | BCQ | BC | DT | StAR | GDT | FCNet | DC | Ours |
| pen-human | 37.5 | 71.5 | 66.9 | 63.9 | 79.5 | 77.9 | 92.5 | 57.7 | **93.8** | 86.6 |
| hammer-human | 4.4 | 1.4 | 0.9 | 1.2 | 3.7 | 3.7 | 5.5 | 1.2 | **43.0** | 31.2 |
| door-human | 9.9 | 4.3 | -0.1 | 2.0 | 14.8 | 1.5 | 20.6 | 0.4 | 22.6 | **25.2** |
| pen-cloned | 39.2 | 37.3 | 50.9 | 37.0 | 75.8 | 33.1 | 86.2 | 50.4 | **98.2** | 89.1 |
| hammer-cloned | 2.1 | 2.1 | 0.4 | 0.6 | 3.0 | 0.3 | 8.9 | 0.2 | 33.8 | **44.6** |
| door-cloned | 0.4 | 1.6 | 0.01 | 0.0 | 16.3 | 0.0 | 19.8 | -0.2 | **23.6** | **23.6** |
| **Average** | 15.6 | 19.7 | 19.8 | 17.5 | 32.2 | 19.4 | 38.9 | 18.3 | **52.5** | 50.1 |
| **Kitchen Tasks** | CQL | IQL | BCQ | BC | DT | StAR | GDT | FCNet | DC | Ours |
| kitchen-complete | 43.8 | 62.5 | 8.1 | **65.0** | 50.8 | 40.8 | 43.8 | 28.0 | 50.0 | 55.0 |
| kitchen-partial | 49.8 | 46.3 | 18.9 | 33.8 | 57.9 | 12.3 | 73.3 | 32.5 | **75.0** | **75.0** |
| **Average** | 46.8 | 54.4 | 13.5 | 49.4 | 54.4 | 26.6 | 58.6 | 30.3 | 62.5 | **65.0** |
| **AntMaze Tasks** | CQL | IQL | BCQ | BC | DT | StAR | GDT | FCNet | DC | Ours |
| antmaze-umaze | 74.0 | **87.5** | 78.9 | 54.6 | 59.2 | 51.3 | 76.0 | 84.0 | 85.0 | 86.9 |
| antmaze-umaze-diverse | **84.0** | 62.2 | 55.0 | 45.6 | 53.0 | 45.6 | 69.0 | 82.0 | 78.5 | **84.0** |
| **Average** | 79.0 | 74.9 | 67.0 | 50.1 | 56.1 | 48.5 | 72.5 | 83.0 | 81.8 | **85.5** |

generalization capabilities. In *Selective Lift*, the model must identify a red cube among various objects and place it onto a green plate.

To train the model, 50 image-based trajectories were collected for the *Lift* and *Place* tasks (with the same set used for both *Place* and *Selective Place* tasks), and 60 trajectories were gathered for the *Selective Lift* task. The latter includes an equal split of trajectories with the red cube on the left and the yellow cube on the right, adding complexity to the model's task execution.

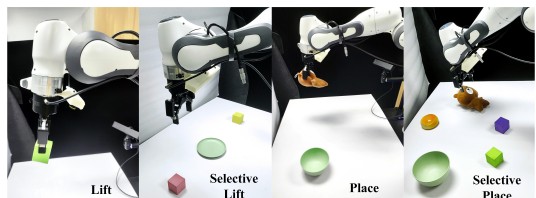

Figure 4: Real-world applications.

## 4.2 Main Results

**Results on D4RL Datasets**    We evaluate DHi across four task domains from the D4RL benchmark: Gym, Adroit, Kitchen, and AntMaze, with results summarized in Table 2. To ensure a fair comparison, we normalize the scores following the protocol established by [9], where a score of 100 represents expert-level performance. We report average scores from five random seeds. DHi demonstrates strong and consistent performance across all domains, surpassing baseline methods in many cases. In the Gym domain, DHi achieves the highest average score of 83.3, indicating its effectiveness in handling a wide variety of tasks. In the Adroit domain, where offline RL struggles with extrapolation errors due to limited human demonstration data, DHi significantly outperforms most baselines and achieves a highly competitive average score. For Kitchen tasks, which require generalization to unseen states and long-term value optimization,

DHi delivers strong performance, indicating its high expressiveness. In the AntMaze domain, which presents challenges due to sparse rewards and sub-optimal trajectories, DHi consistently matches or surpasses existing methods. Overall, these results highlight the robustness of our approach across different domains and task complexities, reinforcing its effectiveness in addressing the challenges posed by the D4RL benchmark.

**Real-World Applications** We further evaluate the performance of DHi and DT on the real-world robotic dataset, measuring their success rates over 10 trials and the inference time for generating 400 actions. As shown in Table 3, DHi achieves a 100% success rate across all tasks, whereas DT struggles with the *Lift* task and performs significantly worse in the more challenging *Selective Lift* task.

Table 3: Performance of real-world experiments. S-Lift and S-Place represent the Selective Lift and Selective Place tasks, respectively.

| Policy | Lift | S-Lift | Place | S-Place | Params/M | Time/s |
|--------|------|--------|-------|---------|----------|--------|
| DT | 0.90 | 0.40 | 1.00 | 1.00 | 9.7 | 2.3 |
| DHi | 1.00 | 1.00 | 1.00 | 1.00 | 6.6 | 1.7 |

Moreover, visualization results (Appendix G and videos in supplementary material) indicate that DHi produces significantly smoother robotic arm trajectories than DT and completes tasks more quickly. Under the same hyperparameter settings, DHi also demonstrates greater efficiency, requiring approximately 32% fewer parameters than DT (6.6M compared to 9.7M) and achieving a 35% improvement in inference speed (1.7s compared to 2.3s). This efficiency gain is primarily due to replacing the self-attention layer with a non-parametric, constant-complexity average pooling operation.

## 4.3 Ablation Study

**The Design of Token Merger** As discussed in the Method section, we evaluate the performance of different Token Mergers across various Token Mixers in the Gym domain (Table 1). Among them, the convolution-based Token Merger consistently achieves the highest average performance. To further analyze its effectiveness, we perform t-SNE visualizations on the same task using convolution-based token merging. Figure 5 shows that before applying the Token Merger, embeddings exhibit a random distribution of states, actions, and RTGs. After applying the Token Merger, the embeddings demonstrate improved local clustering, reinforcing its role in structuring input representations. Combined with the conclusion of Theorem 3.1., these results suggest that the introduction of a Token Merger significantly improves the inter-step modeling consistency. This finding is further supported by evaluations under a more extensive training regime, as detailed in Appendix E.1.

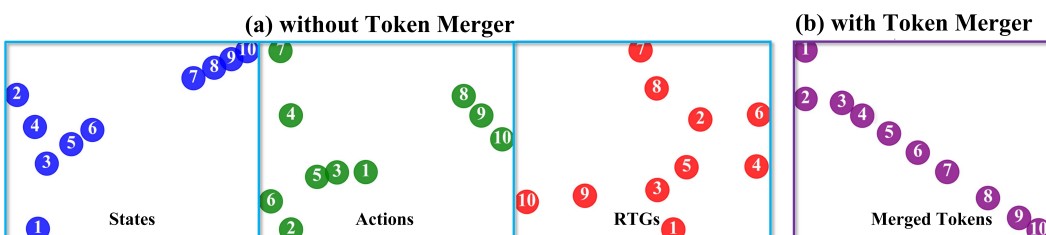

Figure 5: t-SNE visualization of embeddings with/without Token Merger. Post-merger embeddings exhibit structured clustering, indicating enhanced local associations.

**The Size of Pooling Layer** To analyze how pooling sizes influence the model's ability to capture token dependencies, we experiment with pooling sizes of 1, 2, 5, 10, 15 and 20 in the pooling layer. As shown in Appendix E.2, the experimental results show that when the pooling size is set to 1 (equivalent to MLP only), the model's performance significantly decreases due to the lack of contextual history. Smaller pooling sizes (2-10) result in better performance, while excessively large window sizes hinder the modeling of dependencies. We set the pooling size to 2 to align with the Markov property and ensure efficiency.

**Generalization Across Context Lengths and Sparse Reward Settings** We provide an in-depth discussion of our method's robustness to context length and its performance under sparse reward

settings in Appendix F. These analyses further validate the design choices behind DHi, particularly its insensitivity to context size and its resilience in sparse reward conditions.

## 5  Conclusion

In this study, we address the challenges of Offline RL by proposing a novel hierarchical decision modeling framework. Our approach effectively captures state-action-return relationships through intra-step and inter-step modeling. To quantify the model's ability to focus on relevant states, we introduce attention entropy as a metric and reveal its negative correlation with performance. To tackle this issue, we propose the Token Merger to unify representations within each time step and design a parameter-free Token Mixer based on Average Pooling for efficient inter-step dependency modeling. The resulting Decision HiFormer (DHi) outperforms existing baselines on the D4RL benchmark while enhancing computational efficiency. Furthermore, DHi has demonstrated strong generalization capabilities and practical benefits in real-world robotic manipulation tasks, underscoring its applicability beyond simulated environments. A limitation of this study is the lack of evaluation on the transferability of our approach to other RL paradigms, which we leave for future investigation.

## 6  Acknowledgments

This work was supported by the Shanghai Artificial Intelligence Laboratory and the National Natural Science Foundation of China (Grant Nos. 62572313 and 62106139).

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

# A  Pooling with Markov

Given the inherent complexity of deep neural networks, our work builds upon the mathematical framework proposed by [23], focusing on simplified scenarios involving first-order binary Markov data and single-layer pooling token mixer. We conduct a principled analysis of pooling token mixer using Markov chains and further extend the framework to accommodate multi-state Markov processes. Our notation is summarized in Table S1.

Table S1: Mathematical Notations and Definitions

| Notation | Definition | Notation | Definition |
|---|---|---|---|
| $x, y$ | Scalars | $\mathbf{v}$ | Vectors |
| $\mathbf{M}, \mathbf{P}, \mathbf{W}$ | Matrices | $\|\cdot\|$ | $\ell_2$-norm / Frobenius norm |
| $[K]$ | Set $\{1, \ldots, K\}$ | $x_i^j$ | Sequence from $x_i$ to $x_j$ |
| $\sigma(z)$ | Sigmoid function | $\mathrm{ReLU}(z)$ | ReLU function: $\max(0, z)$ |
| $\mathbb{P}(A)$ | Probability of event $A$ | $\mathbb{P}(A \mid B)$ | Probability of $A$ given $B$ |
| $\mathcal{X}$ | Vocabulary, e.g., $\{0, 1\}$ | $N$ | Length of input sequence |
| $\{x_n\}_{n=1}^N$ | Input sequence | $S$ | Vocabulary size (multi-state) |
| $\mathbf{P}_{ij}$ | Transition probability $\mathbb{P}(x_{n+1} = j \mid x_n = i)$ | $\mathbf{P}$ | Transition kernel matrix $(\mathbf{P}_{ij})$ |
| $\boldsymbol{\pi}^{(n)}$ | Probability distribution of state $x_n$ | $\boldsymbol{\pi}^{(1)}$ | Initial state distribution |
| $p, q$ | Switching probabilities $\mathbf{P}_{01} = p, \mathbf{P}_{10} = q$ | $\mathbf{P}(p, q)$ | Binary Markov kernel $\begin{bmatrix} 1-p & p \\ q & 1-q \end{bmatrix}$ |
| $H(Y \mid X)$ | Conditional entropy | $h(p')$ | Binary entropy function |
| $H(x_{n+1} \mid x_n)$ | Markov chain entropy rate | $D_{\mathrm{KL}}(P \parallel Q)$ | Kullback-Leibler divergence |

**Data.**  We assume a binary vocabulary $\mathcal{X} = \{0, 1\}$ and model the input sequence $\{x_n\}_{n=1}^N$ as a first-order Markov chain (i.e., with memory order $m = 1$). In such processes, the probability of the next state depends solely on the current state, independent of any preceding states:

$$\mathbf{P}_{ij} \triangleq \mathbb{P}(x_{n+1} = j \mid x_n = i) = \mathbb{P}(x_{n+1} = j \mid x_1, \ldots, x_n = i)$$

For any $i, j \in \mathcal{X}$, we consider a Markov process governed by the transition kernel $\mathbf{P} = (\mathbf{P}_{ij})$, which determines the system's dynamics. Specifically, if $\boldsymbol{\pi}^{(n)} \in [0, 1]^{|\mathcal{X}|}$ represents the probability distribution of the state $x_n$ at time $n$, then the distribution evolves according to $\boldsymbol{\pi}^{(n+1)} = \boldsymbol{\pi}^{(n)} \cdot \mathbf{P}$.

In this paper, we focus on a specific transition kernel defined over the binary state space:

$$\mathbf{P}(p, q) \triangleq \begin{bmatrix} 1 - p & p \\ q & 1 - q \end{bmatrix},$$

where $p, q \in (0, 1)$ denote the switching probabilities (i.e., $\mathbf{P}_{01} = p$ and $\mathbf{P}_{10} = q$). We define a first-order binary Markov chain $(x_n)_{n \geq 1}$ characterized by the transition kernel $\mathbf{P}(p, q)$ and an initial distribution $\boldsymbol{\pi}^{(1)}$. Formally, we denote this process as $(x_n)_{n \geq 1} \sim (\boldsymbol{\pi}^{(1)}, \mathbf{P}(p, q))$. When the initial distribution $\boldsymbol{\pi}^{(1)}$ is clear from the context, we adopt the simplified notation $(x_{n+1} \mid x_n)_{n \geq 1} \sim \mathbf{P}(p, q)$ to describe the conditional transitions.

The *entropy rate* of a first-order Markov chain, defined as $H(x_{n+1} \mid x_n)$, quantifies the average uncertainty of the next state given the current state. For the binary Markov process considered in this work, the entropy rate is given by

$$H(x_{n+1} \mid x_n) = \frac{q\,h(p) + p\,h(q)}{p + q},$$

where $h(\cdot)$ denotes the binary entropy function.

**Model.** To study a concrete instance of a **Pooling Token Mixer**, we modify the model architecture from [23]. The Self-Attention layer in this base model is substituted with a module comprising an average pooling operation of size $k$ followed by a single-layer LayerNorm. Let $\{x_n\}_{n=1}^N \in \{0, 1\}^N$ denote an input sequence of length $N$. For each $n \in [N]$, the operations within this model are defined as follows:

$$\mathbf{x}_n = x_n \mathbf{e} + \mathbf{p}_n \in \mathbb{R}^d \qquad \text{(Embedding)}$$

$$\mathbf{q}_n = \frac{1}{k} \sum_{i=n-k+1}^n \mathbf{x}_i \in \mathbb{R}^d \qquad \text{(Pooling)}$$

$$\tilde{\mathbf{q}}_n = \boldsymbol{\gamma}_1 \odot \frac{\mathbf{q}_n - f_\mu(\mathbf{q}_n)}{\sqrt{f_\sigma(\mathbf{q}_n)^2 + \epsilon}} + \boldsymbol{\beta}_1 \in \mathbb{R}^d \qquad \text{(LayerNorm)}$$

$$\mathbf{y}_n = \mathbf{x}_n + \tilde{\mathbf{q}}_n \qquad \text{(Shortcut)}$$

$$\mathbf{t}_n = \mathbf{W}_2 \text{ReLU}(\mathbf{W}_1 \mathbf{y}_n) \in \mathbb{R}^d \qquad \text{(Feed-Forward)}$$

$$\mathbf{z}_n = \mathbf{y}_n + \mathbf{t}_n \in \mathbb{R}^d \qquad \text{(Shortcut)}$$

$$s_n = \langle \mathbf{a}, \mathbf{z}_n \rangle + b \in \mathbb{R} \qquad \text{(Linear)}$$

$$f_{\boldsymbol{\theta}}(x_1^n) = \sigma(s_n) \in [0, 1] \qquad \text{(Prediction)}$$

In the above equations, $d$ is the embedding dimension. The term $\mathbf{e} \in \mathbb{R}^d$ is the embedding vector, where specific vectors $\mathbf{e}_1$ (for $x_n = 1$) and $\mathbf{e}_0$ (for $x_n = 0$) are utilized. $\mathbf{p}_n \in \mathbb{R}^d$ represents a trainable positional encoding. For the LayerNorm operation applied after pooling, $\boldsymbol{\gamma}_1, \boldsymbol{\beta}_1 \in \mathbb{R}^d$ are learnable parameters used for scaling and shifting, respectively; $\epsilon > 0$ is a small constant for numerical stability. The functions $f_\mu(\cdot), f_\sigma(\cdot) : \mathbb{R}^d \to \mathbb{R}^d$ compute the mean and standard deviation, respectively, of their input features (here, the pooled features $\mathbf{q}_n$). The feedforward network (FFN) consists of matrices $\mathbf{W}_1 \in \mathbb{R}^{r \times d}$ and $\mathbf{W}_2 \in \mathbb{R}^{d \times r}$, where $r$ is the hidden dimension of the FFN. The final linear layer is characterized by the weight vector $\mathbf{a} \in \mathbb{R}^d$ and scalar bias $b \in \mathbb{R}$, which produce the logits $s_n$. The probability for the symbol $1$ is then computed from these logits using the sigmoid function $\sigma(\cdot)$ as

$$f_{\boldsymbol{\theta}}(x_1^n) \triangleq \mathbb{P}_{\boldsymbol{\theta}}(x_{n+1} = 1 \mid x_1^n) = \sigma(s_n).$$

Note that a single symbol probability suffices, as the vocabulary is binary. The set of all trainable parameters in our model is given by

$$\bar{\boldsymbol{\theta}} \triangleq \left( \mathbf{e}, \{\mathbf{p}_n\}_{n=1}^N, \boldsymbol{\gamma}_1, \boldsymbol{\beta}_1, \mathbf{W}_1, \mathbf{W}_2, \mathbf{a}, b \right).$$

**Loss.** We train the parameters $\bar{\boldsymbol{\theta}}$ using a next-token prediction loss, defined as the cross-entropy loss between the predicted probability $f_{\boldsymbol{\theta}}(x_1^n)$ and the ground truth token $x_{n+1}$ at each position $n \in [N]$. Specifically, the loss function is given by

$$L(\bar{\boldsymbol{\theta}}) \triangleq -\frac{1}{N} \sum_{n \in [N]} \mathbb{E}_{x_{n+1}} \left[ x_{n+1} \log f_{\boldsymbol{\theta}}(x_1^n) + (1 - x_{n+1}) \log \left( 1 - f_{\boldsymbol{\theta}}(x_1^n) \right) \right], \qquad (1)$$

where the expectation is taken over the data distribution of the sequence $\{x_n\}_{n=1}^N$. In practice, this expectation is approximated by the empirical average computed over sequences sampled from the corpus, and stochastic optimizers such as SGD or Adam are used to update the model parameters.

**Objective.** Building on the theoretical framework described above, we investigate the modeling capacity of the aforementioned single-layer pooling-based models (i.e., those utilizing a pooling token mixer). We address the following research question:

*Can single-layer models utilizing a pooling token mixer effectively capture the dynamics of data sequences that exhibit Markovian properties?*

By analyzing the global minimum of the loss function, we derive the theoretical conditions under which these single-layer pooling-based models, when equipped with appropriately designed parameters, can provably model the Markovian dynamics present in input streams with state transition dependencies.

**Lemma 1. (Loss as KL Divergence)** *Let the input sequence $\{x_n\}_{n=1}^N$ be generated by a Markov chain $(\boldsymbol{\pi}(p,q), \mathbf{P}(p,q))$ with fixed $(p,q) \in (0,1)^2$. Let $\boldsymbol{\theta} = (\mathbf{e}_0, \mathbf{e}_1, \{\mathbf{p}_n\}_{n=1}^N, \ldots, b, \mathbf{a}) \in \mathbb{R}^D$ denote the model parameters. The cross-entropy loss $L(\boldsymbol{\theta})$, as defined in Eq. (1), can then be expressed as the sum of the KL divergence between the Markov kernel and the predicted distribution, and the entropy rate of the Markov chain:*

$$L(\boldsymbol{\theta}) = \frac{1}{N} \sum_{n \in [N]} \mathbb{E}_{x_1^n} \Big[ D_{KL}\Big( \mathbb{P}(\cdot \mid x_n) \parallel \mathbb{P}_{\boldsymbol{\theta}}(\cdot \mid x_1^n) \Big) \Big] + H(x_{n+1} \mid x_n),$$

*where $D_{KL}(P \parallel Q)$ denotes the Kullback–Leibler divergence between two distributions $P$ and $Q$, and $H(x_{n+1} \mid x_n)$ is the entropy rate of the Markov chain. The detailed proof is available in [23].*

**Lemma 2. (Gradient Computation)** *Retaining the setup from Lemma 1, let $L(\boldsymbol{\theta})$ be the cross-entropy loss defined in Eq. (1). Then, for any parameter $w \in \boldsymbol{\theta}$, the gradient $\nabla_w L(\boldsymbol{\theta})$ is expressed as:*

$$\nabla_w L(\boldsymbol{\theta}) = -\frac{1}{N} \sum_{n \in [N]} \mathbb{E}_{x_1^{n+1}} \Big[ (x_{n+1} - f_{\boldsymbol{\theta}}(x_1^n)) \nabla_w(s_n) \Big]$$

$$= -\frac{1}{N} \sum_{n \in [N]} \mathbb{E}_{x_1^n} \Big[ (\mathbb{P}(x_{n+1} = 1 | x_n) - f_{\boldsymbol{\theta}}(x_1^n)) \nabla_w(s_n) \Big].$$

*The second equality is derived using the law of total expectation. The detailed proof is available in [23].*

**Theorem 1. (Global Minimum in the Weight-tied Case)** *Let the input sequence $\{x_n\}_{n=1}^N$ be generated by a Markov chain $(\boldsymbol{\pi}(p,q), \mathbf{P}(p,q))$ with fixed $(p,q) \in (0,1)^2$. For the weight-tied case, let $\boldsymbol{\theta} \in \mathbb{R}^{D-d}$ denote the complete set of model parameters. Then, for any $(p,q)$, there exists an explicitly constructed parameter $\boldsymbol{\theta}^\star \in \mathbb{R}^{D-d}$ such that $\boldsymbol{\theta}^\star$ is a global minimum of the population loss $L(\cdot)$ defined in Eq. (1) and its predictions perfectly match the Markov kernel. More precisely, we have:*

*(i)  $L(\boldsymbol{\theta}) \geq L(\boldsymbol{\theta}^\star)$ for all $\boldsymbol{\theta} \in \mathbb{R}^{D-d}$,*

*(ii)  $\mathbb{P}_{\boldsymbol{\theta}^\star}(x_{n+1} = 1 \mid x_1^n) = \mathbb{P}(x_{n+1} = 1 \mid x_n)$,*

*(iii)  $L(\boldsymbol{\theta}^\star) = H(x_{n+1} \mid x_n)$,*

*(iv)  $\nabla L(\boldsymbol{\theta}^\star) = 0$, i.e., $\boldsymbol{\theta}^\star$ is a stationary point.*

*The detailed proof is available in Appendices B.1 and B.2.*

**Corollary 1. (Global Minimum in the Untied Case)** *The weight-tied solution $\boldsymbol{\theta}^* \in \mathbb{R}^{D-d}$ from Theorem 1. can be extended to an untied parameter vector $\bar{\boldsymbol{\theta}}^* \triangleq (\boldsymbol{\theta}^*, \mathbf{a}^*) \in \mathbb{R}^D$ by introducing an independent copy $\mathbf{a}^*$. This construction ensures that the prediction probability under $\bar{\boldsymbol{\theta}}^*$ matches that under $\boldsymbol{\theta}^*$, and thus the true Markov kernel:*

$$\mathbb{P}_{\bar{\boldsymbol{\theta}}^*}(x_{n+1} = 1 \mid x_1^n) = \mathbb{P}_{\boldsymbol{\theta}^*}(x_{n+1} = 1 \mid x_1^n) = \mathbb{P}(x_{n+1} = 1 \mid x_n).$$

*Since this prediction matching (which is property (ii) of Theorem 1. adapted for $\bar{\boldsymbol{\theta}}^*$) is a sufficient condition for global optimality (as established in the proof of Theorem 1., e.g., Appendices B.1 and B.2), it follows that $\bar{\boldsymbol{\theta}}^*$ is a global minimum of the loss $L(\cdot)$ in $\mathbb{R}^D$. Consequently, $\bar{\boldsymbol{\theta}}^*$ also satisfies properties analogous to (i), (iii), and (iv) of Theorem 1., with $\bar{\boldsymbol{\theta}}^*$ replacing $\boldsymbol{\theta}^*$ and the parameter space being $\mathbb{R}^D$.*

**Multi-state Markov Chains.**   Our framework, originally formulated for the binary setting $X = \{0, 1\}$, naturally generalizes to the multi-state setting $X = \{0, 1, \ldots, S-1\}$, where $S$ denotes the state (vocabulary) size. Let the input sequence $(x_n)_{n \geq 1} \sim \mathbb{P}(p)$ be a first-order Markov chain such that, for some $p \in (0,1)$, either $x_{n+1} = x_n$ with probability $1 - p$ or $x_{n+1}$ switches uniformly at random to a different state with probability $\frac{p}{S-1}$. The symmetric kernel $\mathbb{P}(p)$ described earlier (with $p = q$ in the binary case) generalizes naturally to this multi-state setting. For the model architecture, we use the same design as before, except that the embedding and linear layers now incorporate $S$ token embeddings.

# B Proof of Theorem 1

## B.1 Proof of Theorem 1 for $p + q \leq 1$, Weight-tied Case

**Proof.** Assume that $p + q \leq 1$ and that we employ weight tying, i.e., the parameter set is given by

$$\boldsymbol{\theta} = \left(\mathbf{e} = \mathbf{a}, \{\mathbf{p}_n\}_{n=1}^N, \ldots, b\right) \in \mathbb{R}^{D-d}.$$

According to Lemmas 1. and 2., if a set of parameters $\boldsymbol{\theta}$ configures our model such that its output $f_{\boldsymbol{\theta}}(x_1^n)$ perfectly matches the true conditional probability $\mathbb{P}(x_{n+1} = 1 \mid x_n)$, then $\boldsymbol{\theta}$ achieves a global minimum for the loss function $L(\cdot)$. At this minimum, the loss equals the entropy rate of the process, and $\boldsymbol{\theta}$ is also a stationary point.

Consequently, it is sufficient to construct such a parameter set, which we denote $\boldsymbol{\theta}^*$. To inform the design of $\boldsymbol{\theta}^*$, we note that the explicit form of the target Markov kernel is:

$$\mathbb{P}(x_{n+1} = 1 \mid x_n) = x_n (1 - p - q) + p.$$

Therefore, to enforce

$$f_{\boldsymbol{\theta}}(x_1^n) = x_n (1 - p - q) + p,$$

it is sufficient for the model to use only the information from the current symbol $x_n$ and ignore the preceding tokens $x_1^{n-1}$. According to the architecture described in Appendix A, a natural way to achieve this is to set $\mathbf{W}_1 = \mathbf{0}$ and $\boldsymbol{\gamma}_1 = \boldsymbol{\beta}_1 = \mathbf{0}$ in the LayerNorm module. Moreover, by choosing the positional embeddings $\mathbf{p}_n = \mathbf{0}$ for all $n$, we have

$$\mathbf{x}_n = x_n \mathbf{e}, \quad \mathbf{y}_n = \mathbf{x}_n, \quad \mathbf{z}_n = \mathbf{x}_n.$$

The logits are then computed as

$$s_n = \langle \mathbf{e}, \mathbf{z}_n \rangle + b = x_n \|\mathbf{e}\|^2 + b.$$

Since $f_{\boldsymbol{\theta}}(x_1^n) = \sigma(s_n)$ must equal the Markov kernel, we require that

$$\sigma\left(x_n \|\mathbf{e}\|^2 + b\right) = x_n (1 - p - q) + p.$$

Taking the inverse of the sigmoid (i.e., the logit function) yields

$$x_n \|\mathbf{e}\|^2 + b = \log\left(\frac{x_n (1 - p - q) + p}{(1 - p)(1 - x_n) + q\, x_n}\right).$$

By substituting $x_n = 0$ and $x_n = 1$, we obtain:

$$\begin{cases} x_n = 0: & b = \log\left(\dfrac{p}{1 - p}\right), \\ x_n = 1: & \|\mathbf{e}\|^2 + b = \log\left(\dfrac{1 - q}{q}\right). \end{cases}$$

Subtracting the first equation from the second, we obtain

$$\|\mathbf{e}\|^2 = \log\left(\frac{(1 - p)(1 - q)}{pq}\right).$$

Thus, a global minimum $\boldsymbol{\theta}$ must satisfy

$$\|\mathbf{e}\|^2 = \log\left(\frac{(1 - p)(1 - q)}{pq}\right) \quad \text{and} \quad b = \log\left(\frac{p}{1 - p}\right).$$

Note that this choice is well defined since $\frac{(1-p)(1-q)}{pq} > 1$ when $p + q < 1$, implying $\log\left(\frac{(1-p)(1-q)}{pq}\right) > 0$.

Although there are infinitely many solutions $(\mathbf{e}, \mathbf{p}_n, b)$ satisfying the above conditions, a canonical choice for the global minimum $\boldsymbol{\theta} = \boldsymbol{\theta}^*$ is

$$\boldsymbol{\theta}^* = \begin{pmatrix} \mathbf{e} = \mathbf{a} = \mathbf{1}\sqrt{\dfrac{1}{d} \log\left(\dfrac{(1-p)(1-q)}{pq}\right)}, & \{\mathbf{p}_n = \mathbf{0}\}_{n=1}^N, \\ \boldsymbol{\gamma}_1 = \boldsymbol{\beta}_1 = \mathbf{0}, \quad b = \log\left(\dfrac{p}{1-p}\right), & \mathbf{W}_1 = \mathbf{0}, \quad \mathbf{W}_2 = \mathbf{0} \end{pmatrix}$$

where $\mathbf{1} \in \mathbb{R}^{D-d}$ denotes the all-one vector. This completes the explicit construction of $\boldsymbol{\theta}^*$ and hence the proof.

## B.2 Proof of Theorem 1 for $p + q > 1$, Weight-tied Case

**Proof.** We follow a similar strategy as in the $p + q \leq 1$ case by constructing a parameter $\boldsymbol{\theta} \in \mathbb{R}^{D-d}$ such that

$$f_{\boldsymbol{\theta}}(x_1^n) = \mathbb{P}(x_{n+1} = 1 \mid x_n) = x_n (1 - p - q) + p.$$

However, in this case we must utilize the ReLU component of the feed-forward mechanism, unlike the earlier case where we set $\mathbf{W}_2 = \mathbf{0}$. We now describe the construction of $\boldsymbol{\theta}^*$.

First, we set the embedding vectors as $\mathbf{e} = \mathbf{a} = \mathbf{1}$, and the positional encodings as

$$\mathbf{p}_n = -\frac{1}{2}\mathbf{1}, \quad \forall n \geq 1,$$

where $\mathbf{1} \in \mathbb{R}^d$ denotes the all-one vector. Consequently, the embedded input becomes

$$\mathbf{x}_n = x_n \, \mathbf{e} + \mathbf{p}_n = \alpha_n \, \mathbf{1},$$

with

$$\alpha_n = \begin{cases} +\frac{1}{2}, & \text{if } x_n = 1, \\ -\frac{1}{2}, & \text{if } x_n = 0. \end{cases}$$

We also set the LayerNorm parameters to zero:

$$\boldsymbol{\gamma}_1 = \boldsymbol{\beta}_1 = \mathbf{0}.$$

For the feed-forward layer, we choose $\mathbf{W}_1$ and $\mathbf{W}_2$ so that

$$\mathbf{W}_2 \, \text{ReLU}(\mathbf{W}_1 \, \mathbf{y}_n) = \beta_n \, \mathbf{1},$$

where $\beta_n$ is a scalar that may depend on $x_n$. Consequently, the output of the feed-forward block becomes

$$\mathbf{z}_n = \mathbf{x}_n + \mathbf{W}_2 \, \text{ReLU}(\mathbf{W}_1 \, \mathbf{y}_n) = \alpha_n \, \mathbf{1} + \beta_n \, \mathbf{1} = (\alpha_n + \beta_n) \, \mathbf{1}.$$

The logits are then computed as

$$s_n = \langle \mathbf{a}, \mathbf{z}_n \rangle + b = d \, (\alpha_n + \beta_n) + b,$$

since $\mathbf{a} = \mathbf{1}$ and $\langle \mathbf{1}, \mathbf{1} \rangle = d$. As $f_{\boldsymbol{\theta}}(x_1^n) = \sigma(s_n)$ must equal the Markov kernel, we require

$$\sigma\Big(d \, (\alpha_n + \beta_n) + b\Big) = x_n (1 - p - q) + p, \quad x_n \in \{0, 1\}.$$

Taking the logit (i.e., the inverse of $\sigma$) yields

$$d \, (\alpha_n + \beta_n) + b = \log\left(\frac{x_n (1 - p - q) + p}{(1 - p)(1 - x_n) + q \, x_n}\right), \quad x_n \in \{0, 1\}.$$

By substituting $x_n = 0$ and $x_n = 1$ separately and denote the corresponding values of $\beta$ by $\beta_0$ and $\beta_1$, respectively, we obtain:

$$\begin{cases} x_n = 0 : & d(-\frac{1}{2} + \beta_0) + b = \log\left(\dfrac{p}{1 - p}\right), \\ x_n = 1 : & d(\frac{1}{2} + \beta_1) + b = \log\left(\dfrac{1 - q}{q}\right). \end{cases}$$

Subtracting the first equation from the second yields

$$d\Big(1 + \beta_1 - \beta_0\Big) = \log\left(\frac{(1 - p)(1 - q)}{pq}\right). \tag{2}$$

It remains to choose $\beta_0$ and $\beta_1$ so that above equation holds. To this end, consider the design of the feed-forward layer. Let $\mathbf{W}_1 = \omega \, \mathbf{1}\mathbf{1}^\top$ and $\mathbf{W}_2 = -2 \, \mathbf{W}_1^\top$, for some $\omega \in \mathbb{R}_{>0}$. Since $\mathbf{y}_n = \alpha_n \, \mathbf{1}$, we have

$$\mathbf{W}_1 \, \mathbf{y}_n = \omega \, \mathbf{1}\mathbf{1}^\top (\alpha_n \, \mathbf{1}) = \omega \, \alpha_n \, d \, \mathbf{1}.$$

Thus,

$$\text{ReLU}(\mathbf{W}_1 \, \mathbf{y}_n) = \begin{cases} 0, & \text{if } \alpha_n = -\frac{1}{2} \text{ (i.e. } x_n = 0), \\ \frac{\omega d}{2}\mathbf{1}, & \text{if } \alpha_n = \frac{1}{2} \text{ (i.e. } x_n = 1). \end{cases}$$

Then,

$$\mathbf{W}_2 \, \mathrm{ReLU}\big(\mathbf{W}_1 \, \mathbf{y}_n\big) = -2 \, \mathbf{W}_1^\top \, \mathrm{ReLU}\big(\mathbf{W}_1 \, \mathbf{y}_n\big) = \begin{cases} \mathbf{0}, & \text{if } x_n = 0, \\ -\omega^2 \, d^2 \, \mathbf{1}, & \text{if } x_n = 1. \end{cases}$$

Hence, we obtain $\beta_0 = 0$ and $\beta_1 = -\omega^2 \, d^2$. Substituting these into Eq. (2) gives

$$d\Big(1 - \omega^2 \, d^2\Big) = \log\left(\frac{(1-p)(1-q)}{pq}\right).$$

That is,

$$1 - \omega^2 \, d^2 = \frac{1}{d} \, \log\left(\frac{(1-p)(1-q)}{pq}\right).$$

Let $\omega = \omega^*$ be the solution:

$$\omega^* = \sqrt{\frac{1}{d^2}\left(1 - \frac{1}{d} \, \log\left(\frac{(1-p)(1-q)}{pq}\right)\right)}.$$

Next, substitute $\beta_0 = 0$ in the equation for $x_n = 0$ to obtain the bias:

$$d\Big(-\frac{1}{2}\Big) + b = \log\left(\frac{p}{1-p}\right) \quad \implies \quad b = \log\left(\frac{p}{1-p}\right) + \frac{d}{2}.$$

Piecing everything together, we define the canonical global minimizer $\boldsymbol{\theta}^*$ as

$$\boldsymbol{\theta}^* = \begin{pmatrix} \mathbf{e} = \mathbf{a} = \mathbf{1}, & \{\mathbf{p}_n = -\frac{1}{2}\mathbf{1}\}_{n=1}^N, & \boldsymbol{\gamma}_1 = \boldsymbol{\beta}_1 = \mathbf{0}, \\ b = \log\left(\frac{p}{1-p}\right) + \frac{d}{2}, & \mathbf{W}_1 = \omega^* \, \mathbf{1}\mathbf{1}^\top, & \mathbf{W}_2 = -2 \, \mathbf{W}_1^\top \end{pmatrix}$$

This completes the explicit construction of $\boldsymbol{\theta}^*$ and the proof.

## C   Experimental Setup

### C.1   D4RL

In this section, we detail the experimental setup for our Decision HiFormer (DHi) model on the D4RL benchmark [9]. We implement DHi by building upon the official Decision ConvFormer (DC) [15] codebase and incorporate our proposed pooling module. This module performs a **lightweight temporal averaging** operation: for an input sequence (e.g., of merged tokens) $[x_1, x_2, \ldots, x_N]$, it produces an output sequence $[x_1, \frac{x_1+x_2}{2}, \frac{x_2+x_3}{2}, \ldots, \frac{x_{N-1}+x_N}{2}]$. This specific form of pooling, with an effective window of two for the averaged elements, aligns with our goal of efficiently capturing local temporal dependencies from current and preceding timesteps, as discussed for DHi.

Our experiments are conducted on several offline RL tasks from the D4RL benchmark [9], which span diverse challenges including continuous control, dexterous manipulation, and long-horizon navigation with sparse rewards. Specifically, we select representative tasks from the following domains:

- **Gym-MuJoCo**: This domain includes classic continuous control tasks such as HalfCheetah, Hopper, and Walker2d. These tasks generally feature dense reward signals and are used to evaluate the model's ability to learn fundamental motor skills from offline data. The datasets used are *medium* (generated by a medium-performance policy), *medium-replay* (collected from the replay buffer during the training of a medium policy, containing a large volume of suboptimal states), and *medium-expert* (a mixture of trajectories generated by medium and expert policies, providing both generalization challenges and high-quality demonstrations).

- **Adroit**: This domain consists of high-dimensional, fine-grained dexterous manipulation tasks such as Door, Hammer, Pen, and Relocate. These tasks originate from (potentially noisy) human demonstration data and present significant challenges due to action complexity and system dynamics. The datasets used include *human* (trajectories from real human operations, which are natural but may be unstable), *cloned* (data generated by a low-quality policy obtained through behavioral cloning from the 'human' data), and *expert* (data generated from expert policies).

- **AntMaze**: This domain encompasses navigation tasks of varying scales, such as 'umaze', 'medium', and 'large' mazes. The core challenges lie in sparse rewards and long-term temporal dependencies, requiring the agent to perform long-term credit assignment and effective exploration. The datasets used are *play* (non-goal-oriented human teleoperation data, offering broad state coverage but low policy efficiency) and *diverse* (data collected from multiple policies, enhancing state coverage while increasing the learning difficulty).

- **Kitchen**: This domain involves multi-task scenarios where the agent must complete a sequence of four subtasks in a specific order to reach a target configuration, testing generalization and sequential task completion. We use the *complete* dataset (trajectories showing successful, ordered completion of all subtasks) and the *partial* dataset (trajectories with incomplete or out-of-order subtask attempts, representing suboptimal data).

The hyperparameters for Decision HiFormer (DHi) in our D4RL experiments, detailed in Table S2, guide the model's architecture and training. DHi is trained for $10^5$ steps with a batch size of 64. Architecturally, it consists of 3 layers with GELU activation. The context length $K$, defining the length of historical state-action-reward sequences fed into the model, is set to 20. The hidden dimension is optimized from $\{128, 256\}$ based on preliminary experiments. For training, we use the AdamW optimizer with a Cosine learning rate scheduler, selecting an initial learning rate from $\{10^{-4}, 10^{-3}\}$. Regularization includes a weight decay of $10^{-4}$ and gradient clipping at 0.25 to ensure stable training.

Upon completion of training, the model is evaluated online in the respective D4RL task environments. We adopt the standard evaluation protocol from D4RL [9] to quantify performance in these test environments. Specifically, we report the *normalized score*, calculated as:

$$\text{normalized score} = 100 \times \frac{\text{score} - \text{random score}}{\text{expert score} - \text{random score}}.$$

Here, score refers to the agent's raw score, random score is the average score achieved by a random policy, and expert score is the average score achieved by an expert policy for that task. This normalized score typically ranges from 0 to 100, where 0 corresponds to random policy performance and 100 corresponds to expert policy performance. For each task, we report the average of normalized scores over five random seeds, where each seed is evaluated over 100 episodes.

Table S2: Hyperparameter Settings of DHi in D4RL Dataset.

| Hyperparameter | Value |
|---|---|
| Total number of updates | $10^5$ |
| Number of layers | 3 |
| Context length $K$ | 20 |
| Dropout | 0.1 |
| Batch size | 64 |
| Optimizer | AdamW |
| Learning rate scheduler | Cosine Scheduler |
| Learning rate | $\{10^{-4}, 10^{-3}\}$ |
| Weight decay | $10^{-4}$ |
| Gradient norm clip | 0.25 |
| Nonlinearity function | GELU |

We compare our approach with existing SOTA offline RL approaches, each excelling in specific domain tasks. For value-based approaches, we include Implicit Q-learning (IQL) [16], Batch-Constrained deep Q-learning (BCQ) [11], and Conservative Q-Learning (CQL) [17]. For supervised learning approaches, our comparisons encompass Behavioral Cloning (BC), Decision Transformer (DT) [6], StARformer (StAR) [25], Graph Decision Transformer (GDT) [12], Fourier Controller Networks (FCNet) [27] and Decision ConvFormer (DC) [15]. For FCNet and DC, we report the results from the original papers, while results for the other baselines are cited from [13].

Table S3 presents the computation time for one training epoch, GPU memory usage, and the number of parameters. These metrics offer a comparative analysis of the computational efficiency between

Table S3: Comparison of Model Complexity and Training Resources on D4RL.

| Complexity | MLP | DT | DC | DHi (Ours) |
|---|---|---|---|---|
| Training time (s) | 30.4 | 34.8 | 34.2 | 31.9 |
| GPU memory usage | 1.82 GiB | 2.10 GiB | 2.00 GiB | 1.84 GiB |
| Total parameters | 1.84 M | 2.63 M | 1.86 M | 1.84 M |
| Token Mixer parameters | 0.0 K | 789.5 K | 13.8 K | 0.0 K |

MLP vs DT vs. DC vs DHi (Ours), all of which are trained on a single RTX 4090 GPU. It is observed that DHi is more efficient than DT and DC in terms of training time, GPU memory usage, and the number of parameters.

## C.2 Real-World Applications

We further evaluate DHi's inference speed and generalization ability in real-world settings using a 7-DOF Franka Emika Panda robotic arm equipped with a single-joint torque sensor. The model's input consists of RGB images captured by cameras at three positions (right, left, and wrist), as well as 7-dimensional joint position data. The input images are encoded using ResNet50 for visual information, and the joint position features are fused with the visual features through FiLM [24]. These fused features are then passed through a linear layer to transform them into a 512-dimensional state vector, which is input into DHi. The final output is a 10-dimensional action. The hyperparameters of Decision HiFormer and the environment used for real-world evaluation are summarized in Table S4 and Table S5, respectively.

Table S4: Hyperparameter Settings.

| Hyperparameter | Value |
|---|---|
| Total number of updates | $2 \times 10^5$ |
| Number of layers | 3 |
| Hidden dimension | 512 |
| Context length $K$ | 16 |
| Batch size | 8 |
| Optimizer | AdamW |
| Learning rate scheduler | MultiStepLR |
| Learning rate | $10^{-4}$ |
| Nonlinearity function | GELU |

Table S5: Latency Measurement Environment.

| Configuration | Value |
|---|---|
| Warm Up | 10 dummy state inputs |
| CPU | Intel Xeon Platinum 8383C |
| CPU Clock | 2.70 GHz |
| Memory Clock | 3200 MHz |
| OS | Ubuntu 20.04 (Kernel 5.15) |
| Python | 3.10.16 |
| GCC | 9.4.0 |
| Torch Version | 2.1.1 |

# D Supplementary Experiments

## D.1 More Comparison

For extensive comparison, we compare DHi with more baselines: TD3+BC [10], a model-free actor–critic method, utilizing behavior cloning for regularization; RvS [8], a reward-conditioned behavior cloning approach framed as supervised learning; Decision Diffuser (DD) [1], a diffusion-based generative model for iterative trajectory refinement; Waypoint Transformer (WT) [2], a transformer-based sequence-modeling approach using automatically generated subgoals; Decision Mamba (DM) [22], an SSM-based decision-making model with multi-grained architecture. The results are shown in Table S6.

## D.2 Further Validation

To assess whether the experimental patterns observed earlier persist under an extended training regime, we conduct this experiment exclusively on the Hopper-medium dataset, increasing the total number of parameter updates from $2 \times 10^4$ to $1 \times 10^5$, performing evaluations every $10^3$ steps. All other experimental settings remain unchanged.

Table S6: Performance of DHi and more baselines on Gym domains, where dataset abbreviations are: 'medium' as 'm', 'medium-replay' as 'm-r', and 'medium-expert' as 'm-e'. The results are all cited from their original papers.

| Task Name | TD3+BC | RvS | DD | WT | DM | Ours |
|---|---|---|---|---|---|---|
| halfcheetah-m-e | **97.9** | 92.2 | 90.6 | 93.2 | 93.5 | 94.2±1.0 |
| hopper-m-e | **112.2** | 101.7 | 111.8 | 110.9 | 111.9 | 111.7±2.6 |
| walker2d-m-e | 101.1 | 106.0 | 108.8 | 109.6 | **111.6** | 109.6±1.0 |
| halfcheetah-m | 42.8 | 41.6 | **49.1** | 43.0 | 43.8 | 43.4±1.0 |
| hopper-m | **99.5** | 60.2 | 79.3 | 63.1 | 98.5 | 90.1±2.9 |
| walker2d-m | 79.7 | 71.7 | **82.5** | 74.8 | 80.3 | 79.9±2.7 |
| halfcheetah-m-r | **43.3** | 38.0 | 39.3 | 39.7 | 40.8 | 41.5±1.0 |
| hopper-m-r | 31.4 | 73.5 | **100.0** | 88.9 | 89.1 | 97.7±1.9 |
| walker2d-m-r | 25.2 | 60.6 | 75.0 | 67.9 | 79.3 | **81.2**±**8.3** |
| **Average** | 70.3 | 71.7 | 81.8 | 76.8 | 83.2 | **83.3** |

As shown in Table S7, the results obtained by extending the training budget largely corroborate the conclusions drawn from our experiments with a smaller training budget. Specifically, we reconfirm that: (1) the Token Mixer is essential for achieving higher performance (average scores of 81.3-86.2 with mixers vs. 76.6 for MLP-only), though an MLP-only model can perform reasonably well when combined with an effective Token Merger (e.g., 81.2 with the Convolution merger); (2) the performance variation among different Token Mixers remains limited (average scores ranging from 81.3 to 86.2); and (3) Token Mergers consistently improve performance across different Token Mixers and the MLP-only baseline, highlighting their general utility. These findings under a constrained training regime further validate the importance of the Token Mixer component and the synergistic benefits derived from employing Token Mergers.

Table S7: Performance in Hopper-medium task. We report the mean normalized scores obtained across five random seeds.

| Mixer \ Merger | Straightforward | Concatenation | Attention | Convolution | Pooling | Average |
|---|---|---|---|---|---|---|
| None (MLP only) | 64.0 | 76.5 | 79.4 | 81.2 | 82.1 | 76.6 |
| Linear Attention | 76.9 | 89.5 | 79.1 | 94.8 | 90.8 | **86.2** |
| Attention | 67.6 | 78.9 | 85.6 | 89.0 | 85.6 | 81.3 |
| Convolution | 92.5 | 89.2 | 64.4 | 85.1 | 88.2 | 83.9 |
| Pooling | 75.6 | 84.3 | 84.7 | 90.1 | 87.1 | 84.4 |
| **Average** | 75.3 | 83.7 | 78.6 | **88.0** | 86.8 | 82.5 |

## D.3 Standard Deviation

Due to limited space in Table 2 of the main text, the standard deviations for the Gym domain are reported in Table S6, while those for the remaining tasks are shown in Table S8.

Table S8: Standard deviations of DHi across different domains.

| Adroit Domain | | Kitchen Domain | |
|---|---|---|---|
| **Task** | **DHi** | **Task** | **DHi** |
| pen-human | 86.6 ± 11.8 | kitchen-complete | 55.0 ± 4.5 |
| hammer-human | 31.2 ± 5.6 | kitchen-partial | 75.0 ± 5.3 |
| door-human | 25.2 ± 1.9 | | |
| pen-cloned | 89.1 ± 11.1 | **AntMaze Domain** | |
| hammer-cloned | 44.6 ± 9.5 | antmaze-umaze | 86.9 ± 11.9 |
| door-cloned | 23.6 ± 2.1 | antmaze-umaze-diverse | 84.0 ± 13.6 |

# E  Ablation Study

## E.1  The Design of Token Merger

The Token Merger fuses the per-timestep return-to-go (RTG), state, and action into a unified representation, which enables the subsequent encoder layers to focus on modeling inter-timestep dependencies. To investigate the impact of different Token Merger, we compare several Token Merger variants—Attention, Pooling, Concatenation, and Convolution—against a Straightforward baseline that directly feeds separate RTG, state, and action tokens into the model without explicit merging. The comparative results in the Gym domain are summarized in Table S9.

Experimental results show that all Token Merger variants outperform the Straightforward baseline, indicating the importance of explicitly integrating RTG, state, and action. Among the different approaches, the Convolution-based Token Merger consistently achieves the best average performance, as it effectively captures local dependencies between the input tokens and smooths the information fusion. In contrast, the Attention-based variant, known for its strength in global modeling, shows poor performance on this task.

Table S9: Performance of DHi with various Token Merger in Gym domain, where dataset abbreviations are: 'medium' as 'm', 'medium-replay' as 'm-r', and 'medium-expert' as 'm-e'.

| Task Name | Straightforward | Attention | Pooling | Concatenation | Convolution |
|---|---|---|---|---|---|
| halfcheetah-m-e | 89.3 | 92.7 | 88.9 | 90.8 | 94.2 |
| hopper-m-e | 108.9 | 111.7 | 110.8 | 111.0 | 111.7 |
| walker2d-m-e | 108.4 | 96.2 | 108.0 | 107.3 | 109.6 |
| halfcheetah-m | 43.1 | 43.0 | 43.1 | 43.1 | 43.4 |
| hopper-m | 75.6 | 84.7 | 87.1 | 84.3 | 90.1 |
| walker2d-m | 80.1 | 78.8 | 79.5 | 79.0 | 79.9 |
| halfcheetah-m-r | 38.8 | 38.8 | 36.0 | 38.0 | 41.5 |
| hopper-m-r | 83.4 | 92.6 | 95.1 | 97.1 | 97.7 |
| walker2d-m-r | 71.3 | 74.4 | 75.4 | 76.4 | 81.2 |
| **Average** | 77.8 | 79.2 | 80.4 | 80.8 | **83.3** |

## E.2  Analysis of Pooling Size

In this section, we detail our investigation into the impact of varying pooling window sizes on the model's capacity to learn inter-step dependencies. The core of this investigation revolves around the Pooling Module, which executes a lightweight temporal averaging operation. Given an input sequence of tokens $[x_1, x_2, \ldots, x_N]$ and a pooling size $P \geq 1$:

- For the initial $P-1$ positions in the output sequence (i.e., $y_1, \ldots, y_{P-1}$), a simpler averaging or direct mapping is typically employed. For instance, one might set $y_i = \frac{1}{i} \sum_{k=1}^{i} x_k$ for $1 \leq i < P$.
- For all subsequent positions $i \geq P$, the output token $y_i$ is computed as the average of the $P$ input tokens ending at $x_i$: $y_i = \frac{1}{P} \sum_{k=0}^{P-1} x_{i-k}$.

In the specific instance where $P = 2$, as is used in some of our primary configurations, this module's output sequence starts with $y_1 = x_1$, followed by $y_i = \frac{x_{i-1} + x_i}{2}$ for $i > 1$.

We experiment with pooling sizes ($P$) of 1, 2, 5, 10, 15, and 20 for this module. The results, shown in Figure S1(a), indicate a clear trend. Notably, a pooling size of $P = 1$ (which effectively reduces the model to an MLP architecture by negating any temporal averaging beyond the current token if we consider $y_i = x_i$) results in a significant performance drop due to the lack of contextual history integration. In contrast, better results are achieved with smaller pooling windows, particularly for $P$ values between 2 and 10. However, further increasing the window size beyond this range proves detrimental, hindering effective dependency modeling. Based on the design in [15]—which sets filter length to current and previous timesteps per the Markov assumption—we choose a pooling size of $P = 2$ to maintain efficiency and temporal locality.

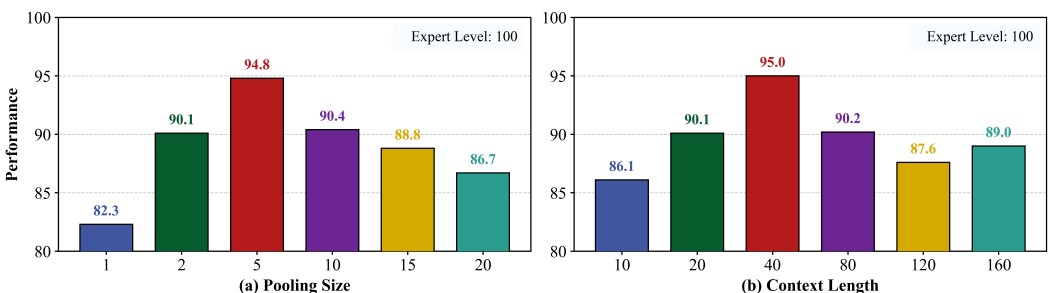

Figure S1: Performance comparison on Hopper-medium task: (a) Impact of varied pooling size; (b) Impact of varied context length.

# F Discussion

## F.1 Context Length

In our experiments on the Hopper-medium task, DHi demonstrates robust performance across both small and large context lengths by replacing the attention layer—traditionally dependent on context length—with a size-2 average pooling layer, as shown in Figure S1(b). For example, the model achieves a performance score of 86.1 with a context length of 10 and 89.0 with a context length of 160. This consistent performance, regardless of the context window length, underscores the model's resilience to variations in context length. Furthermore, by substituting the attention mechanism with a pooling layer, our approach mitigates the potential issue of out-of-distribution generalization, which can occur when there is a mismatch between context lengths during training and testing. In summary, our method effectively reduces sensitivity to context length, ensuring that the model maintains high decision-making quality across a range of settings.

## F.2 Sparse Reward Processing

We assess DHi's performance in a delayed (sparse) reward setting, where rewards are not provided throughout the trajectory but are instead accumulated and given at the final timestep [6, 13]. Table S10 displays the results for both delayed and dense reward conditions in the D4RL-Hopper dataset. The results show that delayed returns minimally affect DHi and DT while CQL collapses, indicating the resilience of sequence modeling approaches in such conditions. Additionally, compared to the straightforward method, DHi with convolution-based Token Merger achieves a notable performance improvement, suggesting that sparse rewards do not affect the local modeling capability of CNNs.

Table S10: Results on the D4RL-Hopper datasets with sparse rewards, where dataset abbreviations are: 'medium' as 'm', 'medium-replay' as 'm-r', and 'medium-expert' as 'm-e'.

| Task Name | Dense Setting | | | Sparse Setting | | |
|---|---|---|---|---|---|---|
| | CQL | DT | DHi | CQL | DT | DHi |
| halfcheetah-m | 49.2 | 42.6 | 43.4 | 1.0 | 42.2 | 43.5 |
| hopper-m | 69.4 | 67.6 | 90.1 | 23.3 | 57.3 | 81.7 |
| walker2d-m | 83.0 | 74.0 | 79.9 | 0.0 | 69.9 | 79.6 |
| halfcheetah-m-r | 45.5 | 36.6 | 41.5 | 7.8 | 33.0 | 40.4 |
| hopper-m-r | 95.0 | 82.7 | 97.7 | 7.7 | 50.8 | 94.3 |
| walker2d-m-r | 77.2 | 66.6 | 81.2 | 3.2 | 51.6 | 74.6 |
| **Average** | 69.9 | 61.7 | 72.3 | 7.2 | 50.8 | 69.0 |

# G  Visualization

## G.1  D4RL Visualization

We observe that the "attention dispersal" problem appears to be a core architectural issue, rather than a direct consequence of trajectory quality. As shown in Figure S2, our visualizations across the medium-expert and medium-replay datasets show similar dispersed attention patterns in the standard DT. Importantly, these visualizations also indicate that this issue is effectively mitigated in each scenario after applying our Token Merger, resulting in a more structured and focused attention maps. These observations support our conclusion that the benefit of our hierarchical structure is not tied to a specific data type, but rather provides a more robust foundation for policy learning across the various datasets of differing qualities found in the D4RL benchmark.

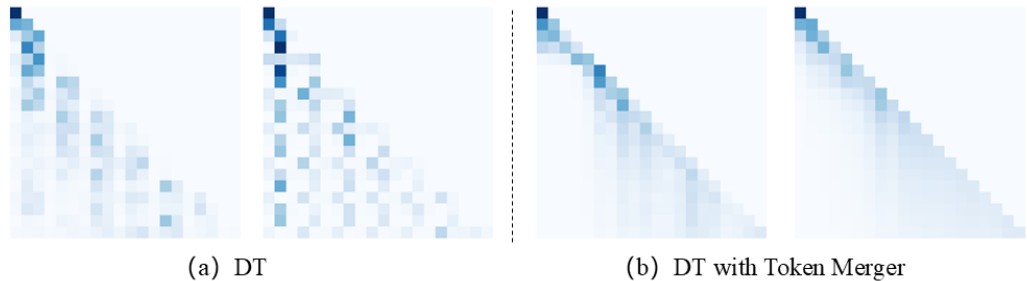

(a) DT                    (b) DT with Token Merger

Figure S2: Attention maps for DT (left) and DT with Token Merger (right) in MuJoCo Hopper-medium-expert and Hopper-medium-replay environments.

## G.2  Real-World Visualization

Visualization of DHi in real-world applications, as shown in Figure S3a, S3b, S3c.

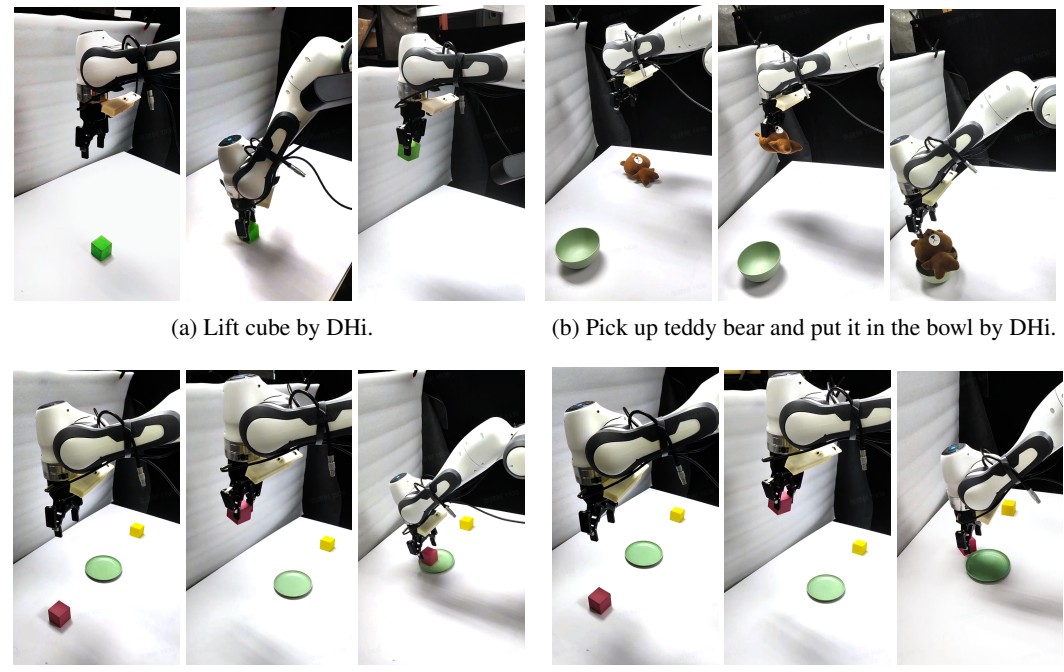

(a) Lift cube by DHi.                    (b) Pick up teddy bear and put it in the bowl by DHi.

(c) Lift red cube and put it on the plate by DHi.        (d) Lift red cube and put it on the plate by DT.

Figure S3: Visualization of DHi and DT in real-world applications

