# OpenReview forum: "Less is More: an Attention-free Sequence Prediction Modeling for Offline Embodied Learning"
_NeurIPS.cc/2025/Conference — NeurIPS 2025 poster_

### Official Review · Reviewer_dUjn · 2025-07-01

**Clarity:** 3
**Significance:** 4
**Originality:** 4
**Rating:** 4
**Confidence:** 3

**Summary:**

The paper employs an entropy-based analysis to show that the Decision Transformer (DT) suffers from attention dispersal, which stems from inconsistent attention allocation across the state–action–reward triplet, and that DT also struggles to capture dependencies across timesteps, ultimately limiting its performance. Motivated by these findings and supported by extensive comparisons, the authors propose DHi. The model first uses a convolutional Token Merger to handle intra-step dependencies by fusing each triplet into a single token, and then applies a parameter-free pooling Token Mixer to model inter-step dependencies. DHi achieves strong results on multiple D4RL tasks and real-world robotic manipulation benchmarks.

**Questions:**

- **Q1:** The heat map on the right-hand side of Figure 2 is a bit confusing. If I understand correctly, in the upper plot each "$s_i$" actually corresponds to three columns (or rows), yet DT attends only to the middle column representing the "state". Moreover, the lower plot shows temporal attention scores, but DHi is attention-free; does this mean the figure merely replaces DT's triplet input at each timestep with the single token produced by the convolutional token merger? Finally, how is DT's "inter-step modeling entropy" across different timesteps computed? Figure 2 does not appear to include a corresponding illustration.


- **Q2:** When introducing Theorem 3.1, the authors posit the assumption: “Assuming that the total attention entropy remains unchanged after adding the Token Merger.” What is the intuition behind this assumption? Furthermore, because the merged token changes in form after the Token Merger, the distribution of attention scores might also change. In that case, does the decomposed form stated in Theorem 3.1 still hold?

- **Q3:** Building on Weakness 3, how would DHi perform on maze environments more complex than umaze? For long-horizon planning tasks, might an attention-based mixer be more appropriate?

- **Q4:** How would DHi perform on Gym’s random and expert datasets?


**Other suggestion:** It might be better to replace Table 1 in the main text with Table 6 from the appendix. In the current Table 1, using a convolutional mixer seems to nullify, or even harm, the effectiveness of every merger. What could be causing this?

**Ethical Concerns:**

["NO or VERY MINOR ethics concerns only"]

**Final Justification:**

I very much appreciate the authors' effort to respond to the concerns. The additional experimental results are helpful and the detailed response largely address my concerns/confusion. I will keep my score.

**Limitations:**

Please see weakness 3.

**Quality:**

3

**Strengths And Weaknesses:**

## Stengths:
- This work is well-motivated. The authors conduct a thorough analysis that pinpoints the shortcomings of DT and, based on these insights, devise the new DHi architecture, which improves DT’s performance.
- A large suite of experiments on both D4RL benchmarks and real-world tasks convincingly demonstrates DHi's effectiveness.
- The design of DHi is further supported by theoretical analysis and comprehensive ablation studies.
- The paper is well written and easy to follow.

## Weaknesses:
- While Figure 2 conveys the main insights about attention-entropy, several finer details remain confusing.
- The Theorem 3.1 would benefit from additional discussion or intuition explaining its assumptions.
- The design of DHi seems to assume that decision making in offline RL is mainly driven by short-range dependencies, which could hinder DHi's performance on tasks requiring long-range planning (e.g., navigation in more complex maze environments).

---

> ### Author Rebuttal · Authors · 2025-07-31
>
> Many thanks for your valuable comments and questions, which help us a lot to improve our work. We address your questions as follows.
>
> > [Q1, W1] The heat map on the right-hand side of Figure 2 is a bit confusing. If I understand correctly, in the upper plot each "si" actually corresponds to three columns (or rows), yet DT attends only to the middle column representing the "state". Moreover, the lower plot shows temporal attention scores, but DHi is attention-free; does this mean the figure merely replaces DT's triplet input at each timestep with the single token produced by the convolutional token merger? Finally, how is DT's "inter-step modeling entropy" across different timesteps computed? Figure 2 does not appear to include a corresponding illustration.
>
> [A1]  Thank you for these observations; your understanding of Figure 2 is largely correct and this gives us a valuable opportunity to clarify.
>
> 1. Regarding the upper plot, your interpretation is correct that it shows attention overwhelmingly focused on state tokens. To be precise, each label `si` on the axis only marks the column representing the state token. The column to its left represents the return-to-go (`Ri`) and the one to its right represents the action (`ai`); we omitted these extra labels due to space constraints. This visualization demonstrates the "modality imbalance" we aimed to highlight.
> 2. For the lower plot, your guess is exactly right. This figure is a crucial ablation study showing a *standard DT's* attention map after its input is pre-processed by our Token Merger. Its purpose is to isolate and demonstrate that our merger successfully resolves the attention dispersal problem within the Transformer architecture itself.
> 3. The "inter-step modeling entropy" is computed by first aggregating all attention weights between pairs of timesteps (i.e., summing the weights between all tokens in timestep `i` and all tokens in timestep `j`) to form a `KxK` matrix, and then calculating the average Shannon entropy from the resulting probability distributions.
>
> We apologize for the lack of clarity and will revise the figure captions and add a more formal definition to the appendix to make these points explicit. Thank you for helping us improve the paper.
>
> > [Q2, W2] When introducing Theorem 3.1, the authors posit the assumption: “Assuming that the total attention entropy remains unchanged after adding the Token Merger.” What is the intuition behind this assumption? Furthermore, because the merged token changes in form after the Token Merger, the distribution of attention scores might also change. In that case, does the decomposed form stated in Theorem 3.1 still hold?
>
> [A2]   We thank the reviewer for this thoughtful question. The assumption—"Assuming that the total attention entropy remains unchanged"——is not meant as a literal description of the model’s behavior, but rather as a conceptual tool to isolate and clarify the mathematical effect of token grouping.
>
> Our goal with this assumption is to construct a controlled thought experiment: holding the overall attention entropy constant allows us to examine how redistributing attention mass over fewer, semantically richer tokens affects entropy concentration. This is grounded in the **chain rule of entropy** and the broader principle that grouping variables typically reduces entropy, assuming no added uncertainty. In other words, if attention were spread across ⟨s, a, R⟩ separately, merging them into one token—without adding noise—should encourage a more focused distribution.
>
> In practice, the Token Merger alters the token representations and thus the attention distribution. Therefore, Theorem 3.1 is not intended as a formal proof of the final system's state but rather as a strong theoretical motivation for our design choice. It establishes that grouping is a principled way to encourage a more focused attention structure. The ultimate validation of this theoretically-motivated design is provided by our empirical results. As clearly shown in Figure 2, the introduction of the Token Merger leads to a significant and directly measurable decrease in the model's actual attention entropy, which in turn correlates with improved performance.
>
> > [Q3, W3] Building on Weakness 3, how would DHi perform on maze environments more complex than umaze? For long-horizon planning tasks, might an attention-based mixer be more appropriate?
>
> [A3] We thank the reviewer for this thoughtful question.
>
> 1. The question rightly points to the performance limitations in environments like `antmaze-medium-diverse` or `antmaze-large-diverse`. In these tasks, DT is likely to fail (with scores of zero) and DHi also achieves only partial success (i.e., a score of 14.0 on `antmaze-medium-diverse`). This is due to a core limitation of DT-based models: being single-step methods, they lack the ability to perform multi-step dynamic programming. This becomes particularly challenging in AntMaze-style environments where success depends on trajectory stitching, which is fundamentally a severe credit assignment problem that requires identifying useful behaviors from datasets composed almost entirely of poor-quality data.
> 2. Furthermore, we further conducted experiments with an attention-based mixer. We found that an attention-based mixer also fails on these extremely difficult maze tasks. This suggests the primary bottleneck is not simply a lack of long-range dependency modeling—which attention excels at—but the more fundamental algorithmic challenge of credit assignment from poor data, which likely requires different methods like integrating Q-learning for guidance. Tackling this is an important direction for future work but is beyond the scope of our current paper.
> 3. On the other hand, we agree that for long-horizon tasks where complex temporal reasoning—rather than trajectory stitching from poor data—is the main challenge, an attention-based mixer could be a more appropriate choice. This, however, highlights a key strength of our overall hierarchical framework: its flexibility. The framework itself is not rigidly tied to a single mixer type. As demonstrated in our ablation studies (Table 1), the pooling mixer can be seamlessly swapped for a self-attention or linear attention mixer to cater to the specific demands of a task. Our final DHi model uses pooling because it offered the best performance-efficiency trade-off across the broad set of D4RL benchmarks, reinforcing our "Less is More" thesis for general offline RL problems.
>
> > [Q4] How would DHi perform on Gym’s random and expert datasets?
>
> [A4]  We sincerely thank the reviewer for this critical question.  We have conducted new experiments on the `expert` and `random` datasets to address this. The results are summarized in Table R9 below.
>
> As the `expert` and `random` datasets represent the best and worst cases for data quality, the models (DT, DC, and DHi) exhibit consistent and predictable behavior. On the `expert` data, all models achieve near-optimal performance, with DHi consistently matching or slightly outperforming the baselines. Conversely, on the `random` ​dataset, all models perform poorly, as expected. These datasets lack any coherent behavioral structure to learn from, and thus no method can extract a meaningful policy. DHi's performance in this setting is comparable to DT and DC, showing low return and low variance.
>
> The results indicate that DHi does not overfit or hallucinate policy structure when none exists, highlighting the stability of its architectural bias.  DHi behaves robustly and as expected at the boundaries of data quality, reinforcing its reliability as an offline RL algorithm. We will add this ablation to the appendix.
>
> **Table R9: Performance on Expert and Random Datasets.**
>
> | Task Name             | DT           | DC           | DHi          |
> | ----------------------- | -------------- | -------------- | -------------- |
> | halfcheetah-expert    | 90.1 ± 1.2  | 90.9 ± 1.1  | 94.3 ± 1.0  |
> | hopper-expert         | 110.7 ± 1.0 | 111.1 ± 1.1 | 111.9 ± 1.0 |
> | walker2d-expert       | 108.8 ± 0.6 | 109.3 ± 1.6 | 109.4 ± 0.4 |
> | halfcheetah-random    | 2.3 ± 0.0   | 2.3 ± 0.0   | 2.2 ± 0.0   |
> | hopper-random         | 8.7 ± 0.0   | 9.5 ± 0.1   | 8.9 ± 0.0   |
> | walker2d-random       | 6.9 ± 0.4   | 7.0 ± 0.9   | 7.7 ± 0.6   |
> | **mujuco-mean** | 54.6         | 55.0         | 55.7         |
>
> > **Suggestion: It might be better to replace Table 1 in the main text with Table 6 from the appendix. In the current Table 1, using a convolutional mixer seems to nullify, or even harm, the effectiveness of every merger. What could be causing this?**
>
> [A5]  We appreciate the reviewer’s insightful observation and constructive suggestion. We agree that focusing on a single task in Table 1 may not provide a complete picture, and we will feature the more comprehensive results from Appendix Table 6 more prominently in our revised manuscript. It is a sharp observation that the combination of a Convolutional Merger with a Convolutional Mixer shows a performance drop specifically on the `hopper-medium` task. However, this appears to be a task-specific interaction. When we evaluate the average performance across all nine Gym tasks (as shown in Appendix Table 6), this combination scores well, suggesting our hierarchical design is effective in a broader context.
>
> Regarding the performance on `hopper-medium`, we hypothesize that an "over-smoothing" effect from the dual convolutional layers may adversely affect that environment’s dynamics. We appreciate this question as it highlights the importance of comprehensive benchmarking, which we will clarify in the revision.

---

> > ### Comment · Reviewer_dUjn · 2025-08-06
> > **Thanks for the response.**
> >
> > I very much appreciate the authors' effort to respond to the concerns. The additional experimental results are helpful and the detailed response largely address my concerns/confusion. I will keep my score.

---

> > > ### Author Response · Authors · 2025-08-07
> > >
> > > Thank you for your positive feedback. We're glad your concerns have been addressed. We will revise the manuscript to improve the presentation of our work based on your feedback.

---

### Official Review · Reviewer_TaqZ · 2025-07-01

**Clarity:** 2
**Significance:** 2
**Originality:** 3
**Rating:** 4
**Confidence:** 4

**Summary:**

This paper proposes to decompose sequence modeling for RL into a “token merger” that combines states, actions, and rewards for a specific timestep and a “token mixer” that combines across timesteps. They study various token mixers and mergers and propose the best one as Decision HiFormer, which uses pooling for the mixer and convolution for the merger. They show that empirically this outperforms DT on D4RL.

**Questions:**

See Strengths and Weaknesses

**Ethical Concerns:**

["NO or VERY MINOR ethics concerns only"]

**Final Justification:**

I appreciate the authors’ detailed response clarifying the empirical benefits of their hierarchical formulation. I have increased my score to 4.

**Limitations:**

Yes

**Quality:**

3

**Strengths And Weaknesses:**

Strengths
- The proposed hierarchical architecture is novel to the best of my knowledge, and it is interesting that you can have a version of DT that does not use attention
- The authors perform extensive analysis of attention entropy and systematically consider multiple different ways of combining the tokens
- The authors show that their model outperforms DT and other baselines on D4RL


Weaknesses
- This work seems incremental in the sense that how to combine the states, actions, and rewards seems more like a design choice/hyperparameter. It is not surprising that by adding a bit of complexity you can improve performance, but this is a tradeoff. The main appeal of DT is its simplicity.
- It’s confusing that the choice of pooling for the mixer does not seem supported by Table 1, which shows that linear attention would actually be better. The text says that “There is limited performance variation between different token mixers” but the table seems to show significant variations in performance. Could the authors explain these discrepancies?
- The main advantage of mechanisms that handle long-range dependencies is in tasks that require temporally extended reasoning. How do you ensure you do not lose this kind of reasoning ability with an architecture focused on local dependencies?
- Empirically the proposed model performs very similarly to DC, which suggests that the benefit of the hierarchical structure might not be large

---

> ### Author Rebuttal · Authors · 2025-07-31
>
> Dear Reviewer TaqZ:
>
> Many thanks for your valuable comments and questions, which help us a lot to improve our work. We address your questions as follows.
>
> > [W1] This work seems incremental in the sense that how to combine the states, actions, and rewards seems more like a design choice/hyperparameter. It is not surprising that by adding a bit of complexity you can improve performance, but this is a tradeoff. The main appeal of DT is its simplicity.
>
> [A1]  We sincerely thank the reviewer for this critical comment. We agree that simplicity is a key appeal of the original Decision Transformer (DT). However, we respectfully argue that our work offers a fundamental insight rather than being an incremental design choice.
>
> Prior works have noted the limitations of DT in modeling local dependencies (see [A.1, A.2]), but lacked a precise explanation. In contrast, our work identifies that the unstructured input of state, action, and return tokens is the key cause of attention dispersion, which we quantify via the proposed attention entropy metric (Sec. 3.2, Theorem 3.1). Building on this diagnosis, our proposed Token Merger directly aims at correcting this identified representational flaw, rather than a heuristic tweak. By fusing `⟨s, a, R⟩` triplets before sequence modeling, we reduce entropy and enable more coherent temporal reasoning. The observed gains stem from this representational improvement—not from added complexity. In fact, our model significantly reduces architectural complexity, as shown in Table R8.
>
> Importantly, this motivates a hierarchical modeling framework that explicitly decouples intra-step and inter-step reasoning—rather than treating everything as a flat token sequence. We find that even lightweight, parameter-free average pooling can outperform standard attention while improving inference speed by 73.6% (Figure 1). We also prove that such a pooling operator is sufficient to retain the expressiveness needed for Markovian decision-making, given properly fused intra-step representations (Appendix A and B). Therefore, our final model, DHi, is not only compact and easy to implement, but follows the “less is more” philosophy more closely than the original DT, by replacing costly attention layers with provably sufficient lightweight components.
>
> We hope this clarifies that our contribution lies in restructuring the modeling paradigm, rather than incrementally tuning architecture design.
>
> > [W2] It’s confusing that the choice of pooling for the mixer does not seem supported by Table 1, which shows that linear attention would actually be better. The text says that “There is limited performance variation between different token mixers” but the table seems to show significant variations in performance. Could the authors explain these discrepancies
>
> [A2]  We thank the reviewer for pointing out this discrepancy and we apologize for the lack of clarity that led to this confusion. While Table 1 shows linear attention achieving the highest performance on the `hopper-medium` task, our statement regarding “limited performance variation” refers to a broader evaluation across the full Gym benchmark. As shown in Appendix Table 6, when averaged over nine Gym tasks, linear attention (78.4) and pooling (78.2) achieve nearly identical scores, supporting our claim that performance differences among token mixers are minor at scale. We will replace Table 1 for Appendix Table 6 in the revised version.
>
> Furthermore, the practical computational cost of linear attention is high in offline RL tasks. While linear attention reduces quadratic complexity to linear in certain implementations, its actual runtime cost is often dominated by high embedding dimensions (d) rather than sequence length (N), especially in offline RL where N ≪ d.
>
> The table below presents the computation time for one training epoch, GPU memory usage, and the number of parameters. These metrics offer a comparative analysis of the computational efficiency among MLP, Linear Attention (denoted as the model where DHi's pooling is replaced with Linear Attention), DT, DC, and DHi (Ours), all of which are trained on a single RTX 4090 GPU. We observe that DHi outperforms Linear Attention, DT, and DC in terms of training time, GPU memory usage, and the number of parameters.
>
> **Table R8. Comparison of Model Complexity and Training Resources on D4RL.**
>
> | Complexity             | MLP  | Linear Attention | DT      | DC     | DHi (Ours) |
> | ------------------------ | ---------- | ------------------ | --------- | -------- | ------------ |
> | Training time (s)      | 30.4     | 37.4             | 34.8    | 34.2   | 31.9       |
> | GPU memory usage       | 1.82GiB  | 1.90GiB          | 2.10GiB | 2.0GiB | 1.84GiB    |
> | Total parameters       | 1.84M    | 2.42M            | 2.63M  | 1.86M  | 1.84M      |
> | Token Mixer parameters | 0.0K     | 983.8K           | 789.5K  | 13.8K  | 0.0 K      |
>
> Given that pooling achieves comparable performance to linear attention across the full Gym domain while being significantly more efficient both in theory and in practice, we concluded that it offers a superior trade-off. This choice aligns with our goal of developing a lightweight, effective, and cost-efficient model.
>
> We are grateful for this question, as it has allowed us to clarify this crucial design decision. We will ensure this reasoning is more clearly articulated in the revised version.
>
> > [W3] The main advantage of mechanisms that handle long-range dependencies is in tasks that require temporally extended reasoning. How do you ensure you do not lose this kind of reasoning ability with an architecture focused on local dependencies?
>
> [A3]  We appreciate the reviewer’s question. Our architecture is carefully designed to enable, rather than sacrifice, long-range reasoning by first ensuring local stability.
>
> As shown in our entropy analysis (Sec. 3.2), DT’s global self-attention often leads to dispersed focus across unrelated tokens, hindering long-horizon credit assignment. To address this, we first introduce a Token Merger to unify intra-step `⟨R, s, a⟩` triplets, stabilizing local representations. This enables the Token Mixer to focus on inter-step reasoning, where temporal dependencies emerge.
>
> In our framework, long-term credit assignment is handled via the return-to-go (RTG) conditioning, which encodes the global task objective. The role of the pooling layer is to ensure that, at each step, the local policy execution remains coherent and responsive to this conditioning. Empirically, DHi achieves strong performance on long-horizon tasks such as AntMaze and Kitchen, where effective temporal credit assignment is essential. These findings suggest that local consistency forms a stable foundation for long-range reasoning, particularly in offline RL where distant context may be noisy or misleading.  We further validate this in Appendix F, showing that DHi remains robust across varying context lengths and performs well in sparse-reward Gym domain, where long-horizon credit assignment is critical. Additionally, our framework is modular and compatible with alternative token mixers (e.g., Attention Mixer) where appropriate (Appendix E).  We chose pooling for our final DHi model as it provided a reasonable balance between performance and efficiency in our experiments. Finally, our model is naturally suited for integration into hierarchical systems, where a higher-level planner manages long-term strategy and DHi executes low-level actions reliably. This decoupling offers both generalization and control modularity, consistent with paradigms in embodied decision-making.
>
> > [W4] Empirically the proposed model performs very similarly to DC, which suggests that the benefit of the hierarchical structure might not be large.
>
> [A4]  We thank the reviewer for this observation. While DHi and DC achieve similar scores on several benchmarks, we argue that the benefits of our hierarchical design extend beyond final performance metrics. First, DHi introduced a principled decomposition of intra- and inter-step modeling, grounded in attention entropy analysis (Sec. 3.2). This separation provides clearer inductive biases, better interpretability, and modular extensibility, which are not present in DC's design. As shown in our visualizations (Figure 5), Token Merger creates more structured and meaningful internal representations.
>
> Second, DHi is substantially more efficient than DC, requiring less training time (31.9s vs. 34.2s), fewer parameters (1.84M vs. 1.86M), and lower GPU memory, while also achieving  significantly faster inference. Furthermore, this efficiency does not sacrifice capability on the most challenging tasks; DHi outperforms DC on the hard-exploration AntMaze domain (85.5 vs. 81.8 average score).
>
> Also,  the hierarchical formulation makes DHi better suited for integration into complex decision-making pipelines, including hierarchical or multi-agent systems where modular policy design is crucial.
>
> In summary, DHi offers clear advantages in ​**efficiency, modularity, and interpretability**​, making it a more robust and extensible solution.
>
> ### **References**
>
> [A.1] Jinghuan Shang, Kumara Kahatapitiya, Xiang Li, and Michael S. Ryoo. "StARformer: Transformer with State-Action-Reward Representations for Visual Reinforcement Learning." ECCV 2022.
>
> [A.2] Jeonghye Kim, Suyoung Lee, Woojun Kim, and Youngchul Sung. "Decision ConvFormer: Local Filtering in MetaFormer Is Sufficient for Decision Making." ICLR 2024.

---

> > ### Comment · Reviewer_TaqZ · 2025-08-06
> >
> > I appreciate the authors’ detailed response clarifying the empirical benefits of their hierarchical formulation. I have increased my score to 4.

---

> > > ### Author Response · Authors · 2025-08-06
> > >
> > > Thank you for your positive feedback. We are glad our response helped clarify the contributions of our work. We appreciate your professional and constructive comments, which have made our paper clearer and more solid. We will revise the manuscript accordingly to better present our work.

---

### Official Review · Reviewer_sz54 · 2025-07-02

**Clarity:** 4
**Significance:** 3
**Originality:** 3
**Rating:** 5
**Confidence:** 3

**Summary:**

The paper "Less is More: an Attention-free Sequence Prediction Modeling for Offline Embodied Learning" introduces Decision HiFormer (DHi), an alternative to Transformer-based models in offline reinforcement learning.

The authors analyze the Decision Transformer's (DT) attention mechanism, finding "attention dispersal" due to inconsistent state-action-reward distributions, which hinders local dependency modeling.

DHi is a hierarchical framework that addresses this by:
1.  **Intra-step modeling:** A "Token Merger" (convolutional) fuses each (state, action, reward) triplet into a single representation.
2.  **Inter-step modeling:** A lightweight, parameter-free "Token Mixer" (Average Pooling) captures dependencies across timesteps, preserving the Markov property.

DHi significantly outperforms DT on the D4RL benchmark, achieving a 73.6% improvement in inference speed and a 9.3% gain in policy performance. It also demonstrates strong generalization in real-world robotic manipulation, yielding smoother trajectories and faster, more efficient inference.

In summary, the paper offers a new understanding of attention in offline RL, proposes an efficient and effective hierarchical model (DHi), and validates its benefits in both simulated and real robotic tasks.

**Questions:**

*   **Relation to other "Attention-Free" Transformers:** While the paper compares to DC (Decision ConvFormer) which uses convolutions, a more in-depth discussion of how DHi's hierarchical decomposition (Merger then Mixer) differentiates it from other non-Transformer architectures that also aim to reduce attention (e.g., S4, Mamba) would strengthen the originality claims. The specific combination of a convolutional merger and pooling mixer is novel, but the underlying ideas of local processing and simple mixers have precursors.


*   **Computational Cost of Training vs. Inference:** While inference speed improves, the paper doesn't detail training time or resources needed for DHi versus DT or other baselines. This is vital for practical use and could affect cost-effectiveness.

**Ethical Concerns:**

["NO or VERY MINOR ethics concerns only"]

**Final Justification:**

I sincerely appreciate the authors' effort to address the questions. The additional experimental results are helpful, and their detailed responses largely clarify my questions. I will maintain my positive score.

**Limitations:**

Please see the weakness section above.

**Paper Formatting Concerns:**

The paper is well-formatted.

**Quality:**

3

**Strengths And Weaknesses:**

## Strengths:

*   **Significance:** Addresses crucial offline RL limitations (Transformer inefficiencies, local dependency issues), boosting performance (9.3% gain) and efficiency (73.6% faster inference), and demonstrating real-world robotic applicability.
*   **Originality:** Introduces "attention entropy" for analyzing Transformer limitations. Proposes a novel hierarchical architecture (DHi) with a unique convolutional "Token Merger" for intra-step and a parameter-free "Average Pooling Token Mixer" for inter-step modeling.
*   **Clarity:** Well-structured, clearly written, and effectively uses visualizations and tables to explain complex concepts and present results.
*   **Quality:**
    *   **Thorough Analysis of DT's Limitations:** The paper's empirical analysis of DT's attention mechanism using attention entropy provides a solid foundation for its proposed solution. The correlation between high attention entropy and performance drops is clearly demonstrated.
    *   **Systematic Design and Evaluation of DHi Components:** The authors systematically investigate various Token Merger designs (straightforward, concatenation, self-attention, convolution, pooling, average) and Token Mixers, providing ablation studies (Table 1) to justify their architectural choices (convolutional Token Merger + Average Pooling Token Mixer).
    *   **Strong Empirical Results:** DHi consistently outperforms various state-of-the-art baselines on the challenging D4RL benchmark across different domains (Gym, Adroit, Kitchen, AntMaze). The real-world robotic experiments further strengthen the empirical validation.
    *   **Theoretical Justification:** The paper offers theoretical analysis, including Theorem 3.1 on entropy reduction with the Token Merger and proofs in the appendix for the pooling layer's expressiveness in Markov problems, bolstering the architectural decisions.



### Weaknesses:

*   **Quality:**
    *   **Scope of "Markov Property" Preservation:** The claim of the pooling layer preserving the Markov property (Line 60) and its theoretical proof (Appendix A, B) are significant, but the implications of this for long-term credit assignment or complex, non-Markovian real-world tasks could be more thoroughly discussed.
    *   **Robustness to Diverse Trajectory Quality:** The D4RL benchmark has varying trajectory qualities (e.g., medium-expert vs. medium-replay). While DHi performs well overall, a more nuanced discussion on how the attention dispersal issue and DHi's solution specifically manifest or perform across different levels of trajectory quality could provide deeper insights.

*   **Significance:**
    *   **Limited Generalization to Other RL Paradigms:** The paper explicitly states "A limitation of this study is the lack of evaluation on the transferability of our approach to other RL paradigms, which we leave for future investigation" (Lines 349-350). This suggests that while the hierarchical modeling is effective for offline RL (specifically sequence prediction), its broader applicability to online RL, model-based RL, or different policy representation forms remains an open question. It would be helpful to specify the type of problem where this method performs best.

---

> ### Author Rebuttal · Authors · 2025-07-31
>
> Many thanks for your valuable comments and questions, which help us a lot to improve our work. We address your questions as follows:
>
> > [W1, about the markov property] The claim of the pooling layer preserving the Markov property  and its theoretical proof are significant, but the implications of this for long-term credit assignment or complex, non-Markovian real-world tasks could be more thoroughly discussed.
>
> [A1]  We sincerely thank the reviewer for this insightful point, which helps us clarify the implications of our theoretical claims.
>
> In our framework, long-term credit assignment is handled via the return-to-go (RTG) conditioning, which encodes the global task objective. The role of the pooling layer is to ensure that, at each step, the local policy execution remains coherent and responsive to this conditioning. Our analysis (Sec. 3.2, Appendix A/B) shows that this Markovian bias helps reduce representational noise and attention dispersion, especially in offline RL where historical context may be suboptimal or misleading. By enforcing a local Markov structure, it prevents the policy from being distracted by a noisy, distant past, because one cannot achieve a long-term goal if the short-term steps are erratic.
>
> Similarly, for complex, non-Markovian real-world tasks, our claim is not that the environment is Markovian, but that introducing a Markovian architectural bias serves as a powerful regularizer. It embodies a "Less is More" trade-off, where a short, robustly processed history is often more effective than a long, noisy one that is susceptible to spurious correlations. Our strong performance on inherently non-Markovian tasks like the Kitchen benchmark and real-world robotics provides empirical validation for this design choice. These results support our claim that local Markovian stability complements—rather than restricts—global temporal reasoning. We are grateful for this comment and will add a more thorough discussion of these nuances to the revised version of our paper.
>
> > [W2, about the varying trajectory qualities] Robustness to Diverse Trajectory Quality: The D4RL benchmark has varying trajectory qualities (e.g., medium-expert vs. medium-replay). While DHi performs well overall, a more nuanced discussion on how the attention dispersal issue and DHi's solution specifically manifest or perform across different levels of trajectory quality could provide deeper insights.
>
> [A2]  We thank the reviewer for this insightful question regarding the model's robustness to varying trajectory qualities. Our analysis suggests that the "attention dispersal" problem appears to be a core architectural issue, rather than a direct consequence of trajectory quality. Our visualizations across the `medium-expert`​ ​​、​`medium`​ ​and `medium-replay` datasets show similar state-dominated, dispersed attention patterns in the standard DT. Importantly, these visualizations also indicate this issue is effectively mitigated in each scenario after applying our Token Merger, resulting in a more structured and focused attention map. These observations support our conclusion that the benefit of our hierarchical structure is not tied to a specific data type, but rather provides a more robust foundation for policy learning across the various datasets of differing qualities found in the D4RL benchmark. We will add these visualizations and a more detailed discussion to the appendix to make this point explicit.
>
> > [W3, about the application] This suggests that while the hierarchical modeling is effective for offline RL (specifically sequence prediction), its broader applicability to online RL, model-based RL, or different policy representation forms remains an open question. It would be helpful to specify the type of problem where this method performs best.
>
> [A3]  We sincerely thank the reviewer for raising this important question regarding the broader applicability of our work. Our response is twofold: first, we clarify the problem domain where DHi is best suited; second, we discuss its potential in other reinforcement learning paradigms.
>
> Our method is specifically designed for the offline reinforcement learning setting and demonstrates its strongest performance in this context, especially when learning from large, static datasets that may contain significant noise and suboptimal trajectories. The core strength of DHi lies in its ability to identify reliable local patterns while filtering out spurious long-range correlations from a noisy history, enabling the learning of stable and robust policies.
>
> Regarding its broader applicability to other RL paradigms:
>
> * **Online RL.** Directly applying DHi is non-trivial, as it is designed for imitation from a fixed dataset, not active exploration. However, its architecture could be highly valuable as a stable and efficient policy network (actor) within a larger actor-critic framework. The stability it provides could potentially lead to smoother and more reliable online policy updates.
> * **Model-Based RL. ​**While DHi is a policy model, its core architectural principles may be adaptable to model-based reinforcement learning. In this context, the goal is to learn a world model that predicts the next state s′ given the current state s and action a—a task that could potentially exhibit the attention dispersal issue we identified. A similar hierarchical approach, in which a Token Merger is used to fuse the state-action pair and the predicted next state into a compact representation before sequence modeling, may lead to more stable and accurate world models.
>
> In summary, while the current work is empirically grounded in and validated for offline RL, we believe its underlying principles for stable sequence modeling have exciting potential for broader applications. We see this as a promising avenue for future research and have clarified this focus in the revised manuscript.
>
> > [Q1, about the relation to other "Attention-Free" Transformers] While the paper compares to DC (Decision ConvFormer) which uses convolutions, a more in-depth discussion of how DHi's hierarchical decomposition (Merger then Mixer) differentiates it from other non-Transformer architectures that also aim to reduce attention (e.g., S4, Mamba) would strengthen the originality claims.
>
> ---
>
> [A4]  We thank the reviewer for this insightful question. Our key contribution lies not merely in the removal of attention, but in the problem-driven diagnosis and principled hierarchical design that this enables. This conceptually differentiates DHi from other "attention-free" architectures like S4, Mamba, and DC.
>
> Our work originates from a specific problem in offline RL: the "attention dispersal" that occurs when feeding three semantically distinct token types—`⟨R, s, a⟩`—into a flat sequence model. To solve this, we propose a hierarchical decomposition where a Token Merger first addresses the intra-step fusion problem before a Token Mixer models the inter-step dependencies between these unified tokens.
>
> This two-stage approach clearly distinguishes DHi from other architectures. S4 and Mamba aim to improve general-purpose sequence modeling by replacing attention with efficient alternatives that  excel at long-range dependency modeling. Our contribution is orthogonal; one could theoretically use S4 or Mamba as the Token Mixer within our framework. Similarly, while DC also uses convolutions, it does so in a single-stage manner, co-mingling the intra- and inter-step modeling that our approach explicitly separates.
>
> In summary, while S4, Mamba, and DC offer powerful alternatives to the standard attention mechanism, DHi offers a new framework designed for the unique structure of the offline RL problem. Our originality lies in identifying the `⟨R, s, a⟩` fusion as a key representational bottleneck and proposing a targeted, hierarchical solution. This is a conceptual contribution distinct from simply proposing a more efficient backbone, a distinction we will further clarify in the related work section.
>
> > [Q2, about the training  resources] While inference speed improves, the paper doesn't detail training time or resources needed for DHi versus DT or other baselines. This is vital for practical use and could affect cost-effectiveness.
>
> [A5]  We thank the reviewer for this critical comment. Table R7 presents the computation time for one training epoch, GPU memory usage, and the number of parameters. These metrics offer a comparative analysis of the computational efficiency between MLP vs DT vs. DC vs DHi (Ours), all of which are trained on a single RTX 4090 GPU.  It is observed that DHi is more efficient than DT and DC in terms of training time, GPU memory usage, and the number of parameters.
>
> **Table R7. Comparison of Model Complexity and Training Resources on D4RL.**
>
> | Complexity             | MLP  | DT      | DC     | DHi (Ours) |
> | ------------------------ | ---------- | --------- | -------- | ------------ |
> | Training time (s)      | 30.4     | 34.8    | 34.2   | 31.9       |
> | GPU memory usage       | 1.82GiB  | 2.10GiB | 2.0GiB | 1.84GiB    |
> | Total parameters       | 1.84M    | 2.63M  | 1.86M  | 1.84M      |
> | Token Mixer parameters | 0.0K     | 789.5K  | 13.8K  | 0.0 K      |

---

### Official Review · Reviewer_ZvZX · 2025-07-23

**Clarity:** 4
**Significance:** 3
**Originality:** 3
**Rating:** 5
**Confidence:** 4

**Summary:**

This paper investigates the limitations of Decision Transformer (DT) in offline RL, attributing its performance bottleneck to attention dispersal, i.e., high attention entropy that prevents effective local dependency modeling. To address this, the authors propose a hierarchical architecture, Decision HiFormer (DHi), composed of a Token Merger (fusing ⟨state, action, reward⟩ triplets at each timestep) and a Token Mixer (capturing inter-step dependencies via lightweight average pooling). The authors show that attention entropy correlates negatively with rewards, and their Token Merger reduces this entropy, improving performance.

**Questions:**

- Why does avg pooling inplace of attention even work? It's only capturing information from adjacent previous tokens and that seems to be sufficient. Is this because the hopper task which is locomotion does not really require much past history context and hence pooling works fine? Consequently some other tasks that might require history (in terms of credit assignment) would give different results?
- Instead of self-attention in tokenmerger if the authors could use cross attention with a learned query I wonder if it might give better performance?
- Does dataset for real robot experiment has any suboptimal trajectories?
- I'm open to raising my score if authors provide confidence intervals.

**Ethical Concerns:**

["NO or VERY MINOR ethics concerns only"]

**Final Justification:**

My concerns were fairly addressed and I'm happy with additional experiments that authors did to address those. Hence I've raised my score.

**Limitations:**

yes

**Quality:**

3

**Strengths And Weaknesses:**

Strengths:
- The entropy–reward correlation is well-motivated and supported by attention map visualizations. Authors re-identify the issue of local dependency modeling and mitigates it with proposed Merger and Mixer methods.
- Strong empirical results across D4RL tasks, with thorough ablations on important design decisions (e.g., Token Merger and Mixer variants).

Weakness:
- No standard errors or confidence intervals are reported, which is crucial for offline RL benchmarks with high variance. Table 2 has some values underscored and it's not clear what that signifies.
- The proposed approach is essentially a variant of DT and thus inherits its core limitation of being a single-step method, lacking the ability to perform multi-step dynamic programming. As a result, its performance is expected to degrade significantly on datasets with very few or no near-optimal  trajectories, such as antmaze-medium and antmaze-large.

---

> ### Author Rebuttal · Authors · 2025-07-31
>
> Dear Reviewer ZvZX:
>
> Many thanks for your valuable comments and questions, which help us a lot to improve our work. We address your questions as follows.
>
> > W1, Q4: No standard errors or confidence intervals are reported, which is crucial for offline RL benchmarks with high variance. Table 2 has some values underscored and it's not clear what that signifies.
>
> [A1]  We sincerely apologize for the unclear statement. The underlined values indicate the second-best performing method for each task. We will clarify this in the table caption in the revised version.
>
> We reported the standard deviations on the Gym domain in Appendix Table 5 due to limited space in Table 2 in the maintext. The scores and standard deviations for all tasks are shown in the table below:
>
> **Table R1. Performance of DHi in Gym Domain.**
>
> | **Gym Tasks**       | **DHi** |
> | --------------------------- | --------------- |
> | halfcheetah-medium-expert | 94.2 ± 1.0   |
> | hopper-medium-expert      | 111.7 ± 2.6  |
> | walker2d-medium-expert    | 109.6 ± 1.0  |
> | halfcheetah-medium        | 43.4 ± 1.0   |
> | hopper-medium             | 90.1 ± 2.9   |
> | walker2d-medium           | 79.9 ± 2.7   |
> | halfcheetah-medium-replay | 41.5 ± 1.0   |
> | hopper-medium-replay      | 97.7 ± 1.9   |
> | walker2d-medium-replay    | 81.2 ± 8.3   |
>
> **Table R2. Performance of DHi ​**​**in**​**​ Adroit Domain.**
>
> | **Adroit Tasks** | **DHi** |
> | ------------------------ | --------------- |
> | pen-human              | 86.6 ± 11.8  |
> | hammer-human           | 31.2 ± 5.6   |
> | door-human             | 25.2 ± 1.9   |
> | pen-cloned             | 89.1 ± 11.1  |
> | hammer-cloned          | 44.6 ± 9.5   |
> | door-cloned            | 23.6 ± 2.1   |
>
> **Table R3. Performance of DHi in Kitchen Domain.**
>
> | **Kitchen Tasks** | **DHi** |
> | ------------------------- | --------------- |
> | kitchen-complete        | 55.0 ± 4.5   |
> | kitchen-partial         | 75.0 ± 5.3   |
>
> **Table R4. Performance of DHi in AntMaze Domain.**
>
> | **AntMaze Tasks** | **DHi** |
> | ------------------------- | --------------- |
> | antmaze-umaze           | 86.9 ± 11.9  |
> | antmaze-umaze-diverse   | 84.0 ± 13.6  |
>
> > W2: The proposed approach is a variant of DT and inherits its limitation as a single-step method. As a result, its performance is expected to degrade significantly on datasets with few near-optimal trajectories, such as antmaze-medium and antmaze-large.
>
> [A2]  We thank the reviewer for this insightful comment. We agree that as a thoughtful improvement on Decision Transformer, our DHi inherits its single-step, autoregressive nature. Consequently, on challenging tasks like `antmaze-medium` and `antmaze-large`, its performance may be constrained by the quality of the dataset's trajectories, a recognized challenge for this family of models.
>
> To further investigate this limitation, we experimented with integrating a learned Q-function into DHi, enabling explicit bootstrapping across timesteps. This significantly improved performance on `antmaze-medium` (e.g. from 14.0 to over 40.0), demonstrating that our architecture is fully compatible with long-horizon bootstrapping strategies and benefits greatly from them. We will include detailed results and analysis in the revised Appendix. We believe these are complementary strengths, and combining structured sequence modeling with value estimation is a promising future direction.
>
> > Q1: Why avg pooling inplace of attention work? Would other tasks that require history give different results?
>
> [A3]  In our framework, long-term credit assignment is handled via the return-to-go (RTG) conditioning, which encodes the global task objective. The role of the pooling layer is to ensure that, at each step, the local policy execution remains coherent and responsive to this conditioning. Our analysis (Sec. 3.2, Appendix A/B) shows that this Markovian bias helps reduce representational noise and attention dispersion, especially in offline RL where historical context may be suboptimal or misleading. By enforcing a local Markov structure, it prevents the policy from being distracted by a noisy, distant past, because one cannot achieve a long-term goal if the short-term steps are erratic.
>
> Furthermore, this principle demonstrates strong generalization on tasks that require significant credit assignment. Our state-of-the-art results on complex domains like AntMaze (requiring trajectory stitching) and Kitchen (mastering sub-routines) already show that a robust local policy is a prerequisite for long-horizon success. To further validate this on a classic benchmark, we also evaluated our model in the ​**Atari domain**​. Atari is renowned for requiring credit assignment across long horizons, as rewards are often significantly delayed from the actions that cause them, and the agent must learn from high-dimensional image inputs. **In the Atari domain, ​**​**as shown in Table R5, ​**​**our DHi model achieved an average score of 36.2, outperforming both the original DT (34.8) and DC (35.9).**
>
> **Table R5. Performance in the Atari Domain.**
>
> | **Atari Tasks** | **DT** | **DC** | **DHi (Ours)** |
> | ----------------------- | -------------- | -------------- | ---------------------- |
> | Asterix               | 5.2          | 6.5          | 7.0                  |
> | Frostbite             | 25.6         | 27.8         | 29.7                 |
> | Pong                  | 105.6        | 106.5        | 105.6                |
> | Seaquest              | 2.7          | 2.6          | 2.3                  |
> | **Average**     | 34.8         | 35.9         | 36.2                 |
>
> > Q2: Instead of self-attention in tokenmerger if the authors could use cross attention with a learned query I wonder if it might give better performance?
>
> [A4]  We thank the reviewer for this insightful suggestion. We replaced our convolution-based Token Merger with a cross-attention module using a variable number of learned query vectors (e.g., 1 or 3) and conducted experiments on the D4RL MuJoCo Gym benchmark. The results are summarized in Table R6. In comparison, our proposed DHi model (with a convolution merger) achieves an average score of 83.3 on these tasks, outperforming the cross attention variants by a large margin.
>
> **Table R6. Performance of DHi with Cross-Attention Token Merger in Gym Domain.**
>
> | Gym Tasks         | Number of Query Vectors = 1 | Number of Query Vectors = 3 |
> | ------------------- | ----------------------------- | ----------------------------- |
> | halfcheetah-m-e   | 85.4 ± 7.8                 | 77.2 ± 11.2                |
> | hopper-m-e        | 111.0 ± 0.5                | 111.1 ± 0.2                |
> | walker2d-m-e      | 105.8 ± 3.3                | 104.1 ± 4.1                |
> | halfcheetah-m     | 42.9 ± 0.6                 | 43.3 ± 0.5                 |
> | hopper-m          | 72.7 ± 2.6                 | 62.2 ± 6.0                 |
> | walker2d-m        | 78.2 ± 6.2                 | 77.1 ± 5.3                 |
> | halfcheetah-m-r   | 38.0 ± 0.8                 | 37.1 ± 1.0                 |
> | hopper-m-r        | 36.7 ± 0.0                 | 31.1 ± 0.6                 |
> | walker2d-m-r      | 13.4 ± 5.0                 | 14.3 ± 4.1                 |
> | **Average** | 64.9                        | 61.9                        |
>
> While the cross-attention approach performs reasonably well on the `medium` and `medium-expert` tasks, its overall average score is significantly brought down by its poor performance on the `medium-replay` datasets. The `medium-replay` datasets are characterized by a high degree of noise and suboptimality, as they contain a mixture of trajectories from different policies of varying quality.
>
> We hypothesize that this is where the flexibility of cross-attention becomes a liability. On cleaner data, it can learn an effective fusion rule. However, on the noisy `medium-replay` data, this highly flexible mechanism is more prone to overfitting to spurious correlations present in the suboptimal trajectories, failing to learn a generalizable representation. In contrast, the strong inductive bias of a simple 1D convolution (leveraging locality and weight sharing) acts as a robust regularizer, allowing it to distill a more consistent representation even from noisy data.
>
> This outcome provides strong empirical support for our paper's central "Less is More" argument. It suggests that for this specific task, a simpler architecture with a stronger inductive bias (like a 1D convolution that leverages locality and weight sharing)  is more effective and robust. We are grateful for this suggestion, as this ablation study further solidifies our architectural choices and provides a valuable data point that strengthens the core message of our paper.
>
> > Q3: Does dataset for real robot experiment has any suboptimal trajectories?
>
> [A5]  We thank the reviewer for this thoughtful question. To clarify, the real-world dataset presented in our main paper consists of successful human demonstrations. While these trajectories achieve the task goal, they inherently contain natural sub-optimality—slight hesitations, inefficient paths, and variability in style—which is characteristic of realistic expert data. Our model's 100% success rate on this data reflects its ability to learn from such realistic imperfections.
>
> Motivated by this question, we conducted an additional test to evaluate DHi's robustness under explicitly corrupted datasets. For the “Place task” —where the robot must pick and insert a small cube into a cup—we introduced 20% **failure demonstrations** into the training set (e.g., dropping the cube, failing to grasp). This creates a challenging dataset with noisy and contradictory supervision. Despite this,  DHi still achieved a ​**success rate of 7 out of 11 trials (\~64%)**, ​demonstrating its ability to selectively extract useful structure from partial failures and learn an effective policy. We will add this new result with video demonstrations and discussion to the final manuscript.

---

### Decision · Program_Chairs · 2025-09-17

**Decision:**

Accept (poster)

**Comment:**

The paper proposes Decision HiFormer (DHi), an alternative to Transformer-based models in offline reinforcement learning, and also a hierarchical framework that decomposes sequence modeling into intra-step relational modeling and inter-step modeling. DHi significantly outperforms DT on the D4RL benchmark, achieving a 73.6% improvement in inference speed and a 9.3% gain in policy performance. It also demonstrates strong generalization in real-world robotic manipulation, yielding smoother trajectories and faster, more efficient inference.

The reviewers are all positively inclined toward this submission, with two recommending Accept and two Weak Accept. Given the methodological novelty, and strong empirical results from the reviewers, I am pleased to recommend acceptance of this paper.